# Post-fertilization transcription initiation in an ancestral LTR retrotransposon drives lineage-specific genomic imprinting of *ZDBF2*

Hisato Kobayashi[1,2]*, Tatsushi Igaki[1], Soichiro Kumamoto[3,4], Keisuke Tanaka[5,6], Tomoya Takashima[1], So I Nagaoka[1], Shunsuke Suzuki[7], Masaaki Hayashi[7], Marilyn B Renfree[8], Manabu Kawahara[9], Shun Saito[9], Toshihiro Kobayashi[10,11], Hiroshi Nagashima[12], Hitomi Matsunari[12], Kazuaki Nakano[12], Ayuko Uchikura[12], Hiroshi Kiyonari[13], Mari Kaneko[13], Hiroo Imai[14], Kazuhiko Nakabayashi[15], Matthew Lorincz[16], Kazuki Kurimoto[1]

[1]Department of Embryology, Nara Medical University, Kashihara, Japan; [2]Department of Medical Genome Science, Dokkyo Medical University, Mibu, Japan; [3]Department of Chemistry and Biochemistry, School of Advanced Science and Engineering, Waseda University, Shinjuku, Japan; [4]Division of Cancer and Senescence Biology, Cancer Research Institute, Kanazawa University, Kanazawa, Japan; [5]Department of Informatics, Tokyo University of Information Sciences, Wakaba, Japan; [6]NODAI Genome Research Center, Tokyo University of Agriculture, Setagaya, Japan; [7]Department of Agriculture, Graduate School of Science and Technology, Shinshu University, Nagano, Japan; [8]School of BioSciences, University of Melbourne, Melbourne, Australia; [9]Laboratory of Animal Genetics and Reproduction, Research Faculty of Agriculture, Hokkaido University, Sapporo, Japan; [10]Division of Mammalian Embryology, Center for Stem Cell Biology and Regenerative Medicine, The Institute of Medical Science, University of Tokyo, Minato, Japan; [11]Center for Genetic Analysis of Behavior, National Institute for Physiological Sciences, Okazaki, Japan; [12]Meiji University International Institute for Bio-Resource Research, Kawasaki, Japan; [13]Laboratory for Animal Resources and Genetic Engineering, RIKEN Center for Biosystems Dynamics Research, Kobe, Japan; [14]Molecular Biology Section, Center for the Evolutionary Origins of Human Behavior, Kyoto University, Inuyama, Japan; [15]Division of Developmental Genomics, Research Institute, National Center for Child Health and Development, Setagaya, Japan; [16]Life Sciences Institute, Department of Medical Genetics, University of British Columbia, Vancouver, Canada

*For correspondence:
hiskobay@naramed-u.ac.jp

Competing interest: The authors declare that no competing interests exist.

## eLife Assessment

The authors analyses describe a novel mechanism by which a retrotransposon-derived LTR may be involved in genomic imprinting and demonstrate imprinting of the ZDBF2 locus in rabbits and Rhesus macaques using allele-specific expression analysis. This imprinting of the ZDBF2 locus correlates with transcription of GPR1-AS orthologs. The accompanying genomic analysis is very well executed allowing for the conclusions reached in the manuscript. The revisions made at the request of the reviewers in this **important** manuscript strengthen the evidence from the genomic analyses, and as a result, the evidence is now **convincing** and will be informative to the genomics and developmental biology communities.

**Abstract** The imprinted gene *ZDBF2* is regulated through a unique mechanism involving a transient paternal transcript in early embryos, rather than persistent gametic DNA methylation. In humans and mice, this transcript—*CMKLR2-AS* (also known as *GPR1-AS*) or the long isoform of *Zdbf2* (*Liz/Zdbf2linc/Platr12*)—arises from the unmethylated paternal allele and initiates secondary epigenetic marks that maintain *ZDBF2* expression. Here, we investigate the evolutionary origin of this mechanism, and show that the first exon of human *GPR1-AS* overlaps with a MER21C long terminal repeat (LTR), a retrotransposon subfamily specific to Boreoeutherian mammals. Comparative analyses revealed that this MER21C insertion occurred in the common ancestor of Euarchontoglires, including primates, rodents, and rabbits. Although not annotated, the first exon of mouse *Liz* displays conserved features with the MER21C-overlapping exon in humans. In rabbit and nonhuman primate placentas, *GPR1-AS* orthologs with LTR-embedded first exons were also identified. In contrast, in non-Euarchontoglire mammals such as cow and tammar wallaby, *ZDBF2* is biallelically expressed, suggesting absence of imprinting. These findings suggest that *ZDBF2* imprinting emerged in Euarchontoglires via MER21C insertion. Together with our prior work on LTR-driven imprinting in oocytes, our findings demonstrate that post-fertilization activation of retrotransposons can also drive lineage-specific acquisition of imprinting.

## Introduction

Genomic imprinting is a well-established biological phenomenon that refers to the differential expression of imprinted genes depending on their parental origin (*Tucci et al., 2019*; *Kobayashi, 2021*). Epigenetic mechanisms, including DNA methylation, histone modification, and noncoding RNA molecules, regulate this process, establishing and maintaining parent-of-origin-specific gene expression patterns. Imprinted genes play vital roles in various biological processes such as fetal growth and development, placental function, metabolism, and behavior. These imprinted genes are often clustered within imprinted domains and regulated by epigenetic marks in specific imprinting control regions (ICRs). The inheritance of parent-of-origin-specific DNA methylation marks from the oocyte or sperm occurs in germline differentially methylated regions (germline DMRs), which are propagated in all somatic tissues of the next generation and function as ICRs (therefore, referred to as canonical imprinting). Imprinting patterns can vary significantly among species. Although some genes are imprinted in only one or a few species, others are conserved in many mammalian lineages. For example, nearly 200 imprinted genes have been identified in mice and humans. However, one comparative analysis suggested that fewer than half of these genes are imprinted in both species, suggesting the existence of many species- or lineage-specific imprinted genes (*Tucci et al., 2019*). The establishment of species-specific imprinting at specific loci can be driven by various evolutionary events, including retrotransposition of genes or transposable elements, which may lead to the acquisition of new genes as imprinted genes or the formation of germline DMRs at CpG-rich regions (CpG islands)(*Kobayashi, 2021*; *Kaneko-Ishino and Ishino, 2022*). These CpG islands, often conserved as orthologous sequences across species, may become germline DMRs in the female germline as a consequence of transcription across the CpG island (*Chotalia et al., 2009*; *Kobayashi et al., 2012a*). Such transcription may be initiated within transposable elements which have integrated nearby. In such cases, the retrotransposition of genes or transposable elements is not directly responsible for the creation of CpG islands themselves but instead facilitates their recruitment into the imprinting system, contributing to the establishment of species-specific imprinting. Indeed, our previous studies have shown that oocyte-specific transcription from upstream promoters, such as integrated LTR retrotransposons, likely played a critical role in the establishment of lineage-specific maternal germline DMRs (*Brind'Amour et al., 2018*; *Bogutz et al., 2019*). An alternative form of imprinting involves the Polycomb marks H3K27me3 and H2AK119ub1, initially deposited in oocytes (*Mei et al., 2021*). Originally described in rodents, such 'non-canonical' imprints are maternally inherited and in turn silence associated genes during preimplantation. Subsequently, these Polycomb marks are converted into DNA methylation over LTR elements, thus maintaining silencing of maternal alleles in the extraembryonic lineage (*Hanna et al., 2019*; *Richard Albert et al., 2023*). In addition, interspecies differences in the regulators of ICRs, such as *ZFP57* and *ZNF445*, as well as species-specific orthologs of DNA methyltransferases, including *DNMT3C*, are potential species-specific features: *ZFP57* and *ZNF445* cooperatively maintain ICR methylation, with *ZNF445* being more dominant in humans and *ZFP57* in mice;

and *DNMT3C* is crucial for a paternally methylated DMR and appears to be specific to Muroidea. These features might also have contributed to the establishment of species-specific imprinted genes (*Barau et al., 2016*; *Takahashi et al., 2019*). However, these reports alone have not been sufficient to propose a comprehensive model for the emergence of all species-specific imprinted genes.

*Zinc finger DBF-type containing 2* (*ZDBF2*), a paternally expressed gene located on human chromosome 2q37.1, has been reported to regulate neonatal feeding behavior and growth in mice, although its molecular function remains unknown (*Kobayashi et al., 2009*; *Glaser et al., 2022*). There are at least three imprinted genes, *ZDBF2*, *GPR1* (alternatively named *CMKLR1*), and *ADAM23* around this locus (*Hiura et al., 2010*; *Morcos et al., 2011*). Recently, the *Zdbf2* gene was also shown to be paternally expressed in the rat, another member of the Muridae family (*Richard Albert et al., 2023*; *Supplementary file 1*). Studies have shown that *ZDBF2* imprinting is regulated by two types of imprinted DMRs: a germline DMR located in the intron of the *GPR1* gene and a somatic (secondary) imprinted DMR upstream of *ZDBF2* (*Kobayashi et al., 2012b*; *Kobayashi et al., 2013*; *Duffié et al., 2014*). The germline DMR is methylated on the maternal allele, leading to silencing of *GPR1* antisense RNA (*GPR1-AS*, alternatively named *CMKLR1-AS*) expression on that allele. In contrast, the paternal allele is hypomethylated, allowing for the expression of *GPR1-AS*, which is transcribed in the reverse direction of *GPR1* (in the same direction as *ZDBF2*) from the germline DMR. Although it is not known whether *GPR1-AS* encodes a protein, a fusion transcript of *Gpr1-as* (*Platr12*) and *Zdbf2* (alternatively named *Zdbf2linc* or *Liz*; a long isoform of *Zdbf2*) has been identified in mice (*Kobayashi et al., 2012b*; *Duffié et al., 2014*). Notably, this fusion transcript exhibited an imprinted expression pattern similar to that of human *GPR1-AS* (*Supplementary file 1*). *Liz* transcription counteracts the H3K27me3-mediated repression of *Zdbf2* by promoting the deposition of antagonistic DNA methylation at the secondary DMR (*Greenberg et al., 2017*). Germline DMRs that function as ICRs generally maintain uniparental methylation throughout development; however, differential methylation of the germline DMR is no longer maintained at this locus after implantation because the paternally derived allele also becomes methylated after implantation. Thus, the secondary DMR, rather than germline DMR, is thought to maintain the imprinted status of this locus in somatic cell lineages. The unique regulatory mechanism responsible for imprinting at the *ZDBF2* locus appears to be conserved between humans and mice; however, whether this conservation extends to other mammals remains unexplored.

Here, we focus on *GPR1-AS/Liz*, which is critical for the imprinting of *Zdbf2* in mice, and identified orthologous transcripts of *GPR1-AS* in primates and rabbits using RNA-seq-based transcript analysis. Strikingly, the first exon of these orthologs overlaps with a common LTR retrotransposon, suggesting that LTR insertion in a common ancestor led to the establishment of *ZDBF2* imprinting. In support of this hypothesis, *ZDBF2* expression is not imprinted in a mammalian outgroup to the Euarchontoglires that lack this LTR, including cattle and tammar wallabies. Taken together, our findings indicate that in the branch of the mammalian lineage that shows imprinting of *ZDBF2*, the key evolutionary event was the insertion at this locus of an LTR-derived sequence that becomes active only after fertilization.

## Results

### Detection of *GPR1-AS* orthologs using RNA-seq data sets

*GPR1-AS* and *Liz* expression have been observed in placental tissues, blastocyst-to-gastrulation embryos, and/or ES cells in humans and mice (*Kobayashi et al., 2009*; *Kobayashi et al., 2012b*; *Kobayashi et al., 2013*; *Duffié et al., 2014*). Public human tissue RNA-seq data sets show that *GPR1-AS* transcription is detectable in the placenta, albeit at a relatively low level (RPKM value of around 0.45—1.8), but is significantly suppressed in other tissues (*Figure 1—figure supplement 1A*; *Fagerberg et al., 2014*; *Duff et al., 2015*). To determine whether *GPR1-AS* orthologs are present across different mammalian species, we performed transcript prediction analysis using short-read RNA-seq data from the placenta or specific extra-embryonic tissues of Eutheria (placental mammals) and Metatheria (marsupials).

First, we downloaded the placental RNA-seq data from 15 mammals: humans, bonobos, baboons, mice, golden hamsters, rabbits, pigs, cattle, sheep, horses, dogs, bats, elephants, armadillos, and opossums (*Armstrong et al., 2017*; *Mika et al., 2022*). These datasets were generated using conventional non-directional RNA-seq (also known as non-stranded RNA-seq), which does not retain information regarding the genomic element orientation of the sequenced transcripts. In addition, library

preparation methods and the amount of data varied significantly among the datasets, ranging from millions to tens of millions (*Supplementary file 2*). Although transcriptional analysis of these datasets identified transcripts such as *GPR1-AS* in human and baboon placentas, the structures of these predicted transcripts were so fragmented that their full-length forms could not be robustly determined (*Figure 1—figure supplement 1B-C*). Furthermore, we did not observe such a transcript at the *Gpr1-Zdbf2* locus in the mouse placental data, where *Liz* is known to be expressed. This suggests that conventional RNA-seq datasets may not be sufficient to accurately detect *GPR1-AS* in different species.

Next, we downloaded directional RNA-seq data (also known as strand-specific or stranded RNA-seq, which can identify the direction of transcription) from the human placenta, rhesus macaque trophoblast stem cells, and mouse embryonic placenta (*Necsulea et al., 2014*; *Rosenkrantz et al., 2021*). In each of these deeply sequenced datasets, which include a minimum of 50 million reads, transcripts originating from the *GPR1* intron were detected in the opposite direction to *GPR1*, extending towards the *ZDBF2* gene (i.e. *GPR1-AS* in humans and rhesus macaques, and *Liz* in mice) (*Figure 1*). Thus, *GPR1-AS* transcripts can be identified in placental tissues using deep directional RNA-seq data.

To identify *GPR1-AS*-like transcripts in other mammalian species, we generated deep directional RNA-seq datasets from the whole placenta of chimpanzees and the trophectoderm (TE) of rabbit, bovine, pig, and opossum post-gastrulation embryos, which yielded a total of 50–100 million reads (*Supplementary file 2*). Among these datasets, *GPR1-AS* orthologs were identified in chimpanzees and rabbits but were not observed in cows, pigs, or opossums (*Figure 2*). Additionally, we performed directional RNA-seq of tissues from the embryonic proper (embryonic disc: ED) of multiple individual animal embryos but did not identify any *GPR1-AS*-like transcripts in any of these samples (*Figure 2—figure supplement 1*). Thus, rabbit *GPR1-AS* is expressed only in placental lineages. Identification of *GPR1-AS* in several primate species as well as mice and rabbits but not in cows, pigs, or opossums indicates that the regulatory region driving *GPR1-AS* transcripts likely originated in the common ancestor of the Euarchontoglires group.

## Origin of *ZDBF2* imprinting and *GPR1-AS* transcript

Since *GPR1-AS/Liz* is essential for *ZDBF2* imprinting in mice, it is assumed that *ZDBF2* imprinting system via *GPR1-AS* transcription is not established in other eutherians and marsupials that lack *GPR1-AS* (*Greenberg et al., 2017*). To test this hypothesis, we performed allele-specific expression analysis of the *ZDBF2* gene in embryonic, fetal, and adult tissues of tammar wallabies and cattle, which are outside the Euarchontoglires group. We identified available SNPs at the 3'-UTR of *ZDBF2* in each of these mammals, and Sanger-sequencing-based polymorphism analysis showed the expression of both *ZDBF2* alleles in all bovine and wallaby tissues analyzed (*Figure 3A–B*; *Supplementary file 1*). Similar analyses were performed using blood samples from rhesus macaques and rabbits, in which *GPR1-AS* was identified. Consistent with our findings in humans and mice, we observed paternal allele-specific expression of *ZDBF2* in these species (*Figure 3C–D*), in clear contrast to the bi-allelic expression observed in the embryonic, fetal, and adult tissues of tammar wallabies and cattle.

To investigate a potential initiating mechanism in the germline for *ZDBF2* imprinting, which is established prior to the secondary DMR, we examined the presence of a germline DMR in both species with and without *ZDBF2* imprinting and *GPR1-AS* expression. Analysis of published whole-genome bisulfite sequencing data from rhesus macaque oocytes and sperm revealed DMRs between both germ cells, including an oocyte-methylated germline DMR at the first exon of *GPR1-AS* (*Figure 3—figure supplement 1A*; *Gao et al., 2017*). This finding aligns with previous observations in humans and mice (*Kobayashi et al., 2012b*; *Kobayashi et al., 2013*; *Duffié et al., 2014*). In contrast, analysis of published whole-genome bisulfite sequencing data for porcine and bovine oocytes and sperm (*Ivanova et al., 2020*) showed no oocyte-methylated germline DMRs in the *GPR1* intragenic region, where *GPR1-AS* is transcribed from an intron of *GPR1* in humans and rhesus macaques (*Figure 3—figure supplement 1B*). Taken together, these results support the hypothesis that *ZDBF2* imprinting is restricted to mammals within the Euarchontoglires group.

Notably, the first exon of all *GPR1-AS* orthologs, except for mouse *Liz,* overlapped with an annotated MER21C element, a member of the LTR retrotransposon subfamily (*Figures 1 and 2*, *Figure 1—figure supplement 1C*). While MER21C retrotransposons are broadly present in Boroeutherians, among the mammals tested, only primate and rabbit genomes have MER21C in the *GPR1* intron,

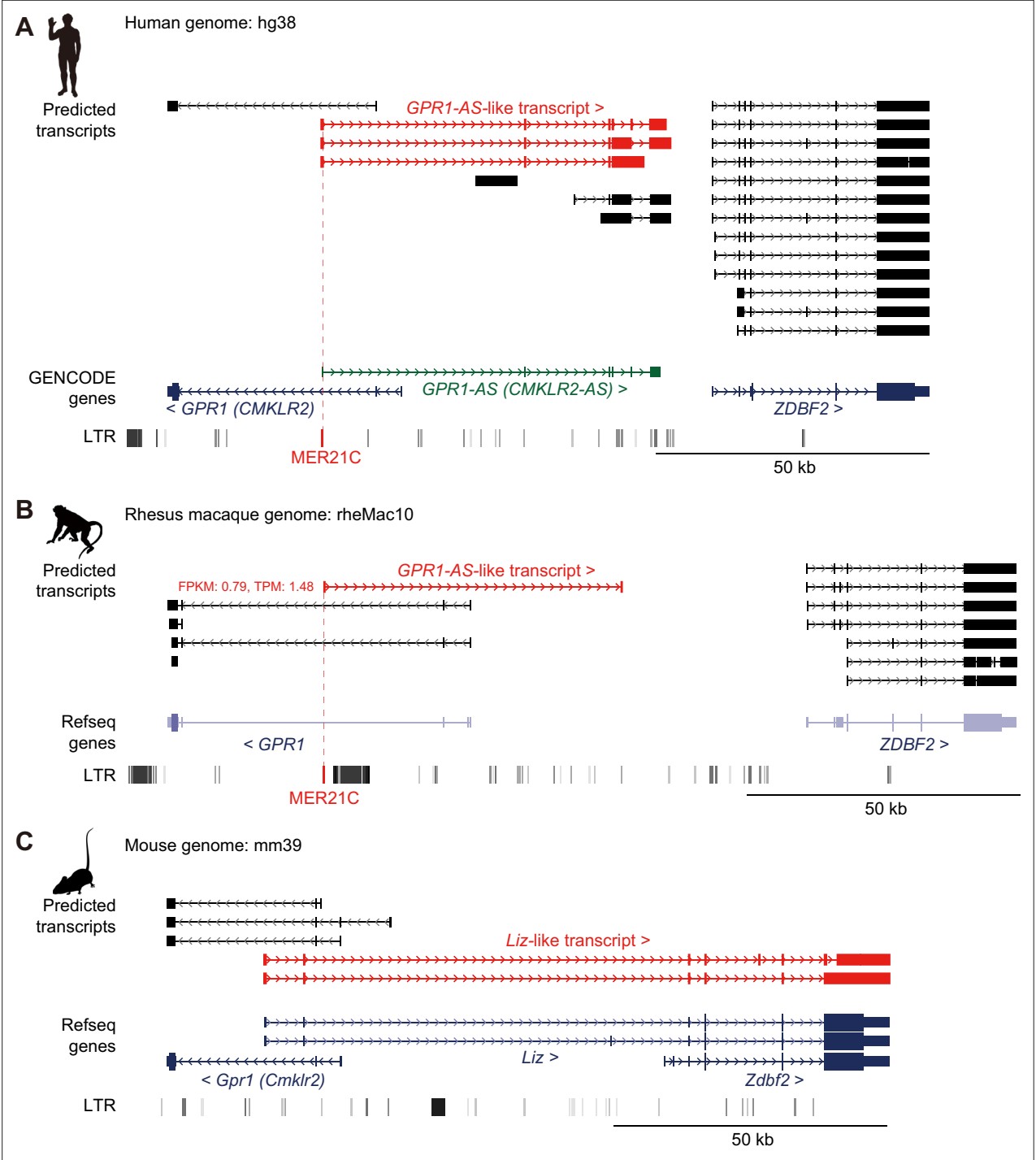

**Figure 1.** Identification of *GPR1-AS* orthologs from public placental transcriptomes. UCSC Genome Browser screenshots of the *GPR1-ZDBF2* locus in humans (**A**), rhesus macaques (**B**), and mice (**C**). Predicted transcripts were generated using public directional placental RNA-seq datasets (accession numbers: SRR12363247 for humans, SRR1236168 for rhesus macaques, and SRR943345 for mice) using the Hisat2-StringTie2 programs. Genes annotated from GENCODE or RefSeq databases and long terminal repeat (LTR) retrotransposon positions from UCSC Genome Browser RepeatMasker tracks are also displayed. Among the gene lists, only the human reference genome includes an annotation for *GPR1-AS* (highlighted in green). *GPR1-AS*-like transcripts and MER21C retrotransposons are highlighted in red. Animal silhouettes were obtained from PhyloPic. Animal silhouettes were obtained from PhyloPic (mouse silhouette by Katy Lawler, available under a CC BY 4.0 license).

The online version of this article includes the following figure supplement(s) for figure 1:

**Figure supplement 1.** Identification of *GPR1-AS* orthologs using public and non-directional RNA-seq data.

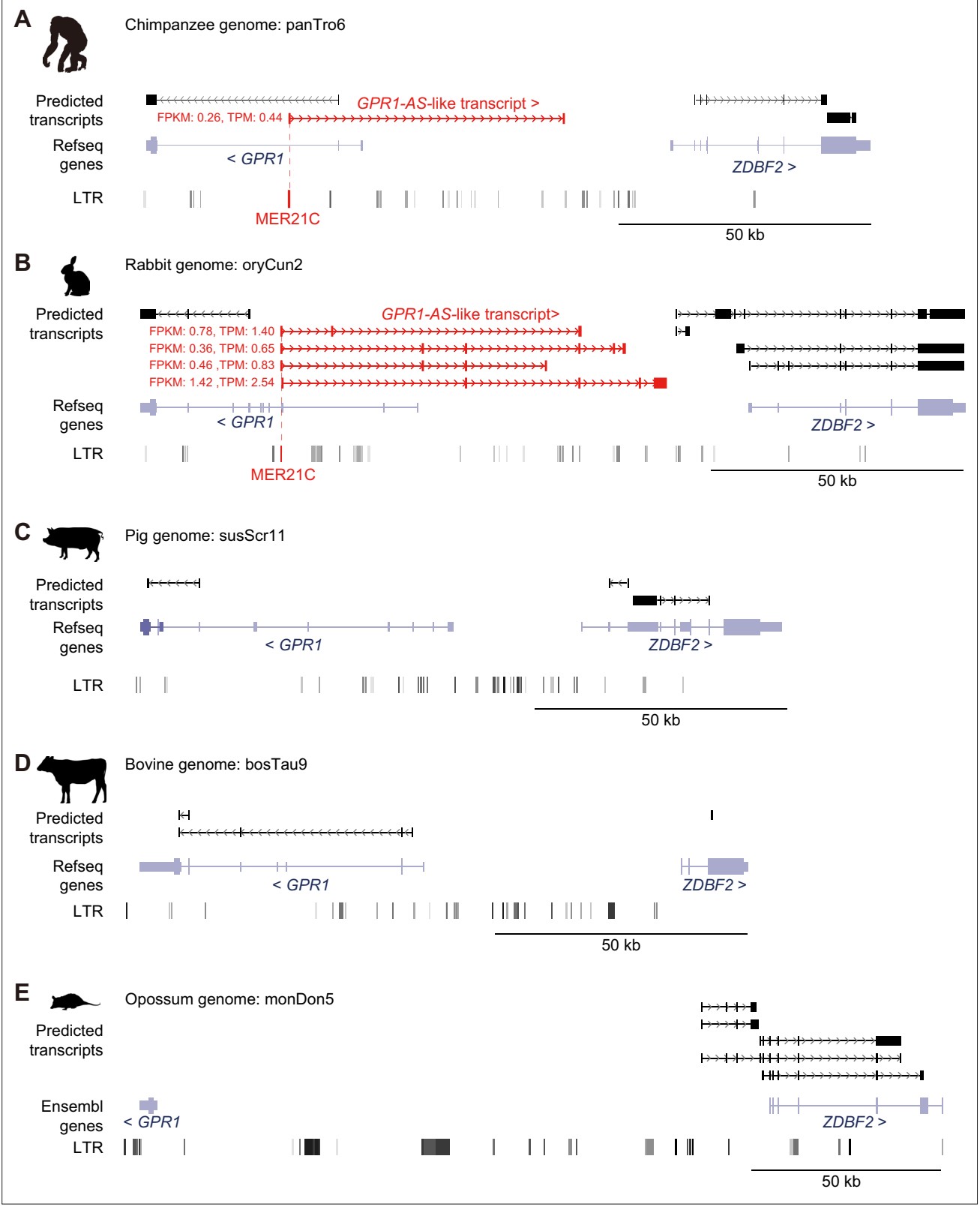

**Figure 2.** Identification of *GPR1-AS* orthologs from original placental and extra-embryonic transcriptomes. Predicted transcripts were generated from placental and extra-embryonic directional RNA-seq datasets of chimpanzee (**A**), rabbit (**B**), pig (**C**), cow (**D**), and opossum (**E**) with the Hisat2-StringTie2 programs. Genes annotated from RefSeq or Ensembl databases and their long terminal repeat (LTR) positions are also shown. MER21C

*Figure 2 continued on next page*

*Figure 2 continued*

retrotransposons, *GPR1-AS*-like transcripts, and their fragments per kilobase million (FPKM) and transcripts per kilobase million (TPM) values are highlighted in red. Animal silhouettes were obtained from PhyloPic (opossum silhouette by Sarah Werning, available under a CC BY 3.0 license).

The online version of this article includes the following figure supplement(s) for figure 2:

**Figure supplement 1.** Search for *GPR1-AS* orthologs from embryonic transcriptomes.

where the first exon of *GPR1-AS* is located. Therefore, we compared the LTR sequence location information of the syntenic region of the *GPR1* and *GPR1-AS* loci to profile MER21C insertion sites in several mammalian genomes, excluding marsupials. Among eutherians, we found the insertion of MER21C (or MER21B retrotransposons that showed high similarity to MER21C) in the introns of *GPR1* in five groups of Euarchontoglires, including primates, colugos, treeshrews, rabbits, and rodents, with the exception of mouse, rat, and hamster (***Figure 4***). As the default parameters of RepeatMasker may not detect degenerate LTR sequences, we reanalyzed these three rodent species using less stringent RepeatMasker parameters but again failed to reveal the presence of an MER21C insertion in the relevant regions (***Figure 4—figure supplement 1***). However, multiple genome alignments from UCSC/Cactus track indicated that the MER21C-derived sequence on the human *GPR1-AS* is likely conserved in mammals belonging to Euarchontoglires, including mouse, rat, and hamster (***Figure 4—figure supplement 2***; ***Armstrong et al., 2020***; ***Zoonomia Consortium, 2020***), albeit with a high level of degeneracy in the latter.

To analyze this syntenic region in greater detail, we compared MER21C sequences that overlapped with the first exon of *GPR1-AS* in each primate (345 bp, 339 bp, and 506 bp from humans, chimpanzees, and rhesus macaques, respectively) as well as rabbits (218 bp), with the first exon of *Liz* in mice (317 bp), and the common sequence of MER21C (938 bp). Pairwise alignment revealed high identities between primate sequences (identity: 71–72%) and the consensus MER21C sequence compared to rabbit (60%) and mouse sequences (46%) (***Figure 5—figure supplement 1A***). The sequence identity in mice is so low that it cannot be distinguished from other LTRs or retrotransposons, which explains why it was not classified as MER21C by RepeatMasker analysis (***Figure 5—figure supplement 1B–C***). Multiple sequence alignments and phylogenetic tree reconstruction reveal that the rabbit and mouse sequences deviate to a greater extent from the consensus MER21C sequence than the cluster formed by the primate sequences, indicating that a greater number of mutations have accumulated in the non-primate lineage of Euarchontoglires since their divergence from primates (***Figure 5A***). This observation aligns with the shorter generation time of lagomorphs and rodents compared to primates (***Wu and Li, 1985***; ***Huttley et al., 2007***).

To evaluate whether the putative degenerate MER21C-derived regulatory region driving mouse *Liz* expression confers transcriptional activity in the mouse comparable to that of the MER21C element that drives human *GPR1-AS expression*, we performed a dual reporter assay in the human cell line HEK293T. Strikingly, the first exon of mouse *Liz* exhibits promoter activity greater than that of the human *GPR1-AS* promoter, despite the relatively low sequence similarity between the *Liz* first exon and the consensus MER21C sequence (***Figure 5—figure supplement 2***). Taken together, these findings suggest that despite substantial sequence divergence, the functional elements responsible for initiating transcription of Liz in the mouse are derived from the same MER21C relic annotated in the human locus and can drive robust expression in human cells.

If this is indeed the case, then specific transcription factor binding motifs are likely shared between species in this regulatory region. To identify common cis-motifs, we compared these sequences with known transcription factor-binding motifs using the TOMTOM program in the MEME suite. Through this analysis, we found a common region that contained an ETS family transcription factor (ELF1 and ELF2) binding site, which had significant matches with E-values <0.01 and q-values <0.01 (***Figure 5B and C***). Additionally, we identified the TFAP2 and ZSCAN4 binding motifs (with p-values = $1.58×10^{-5}$ and $7.24×10^{-7}$) from the JASPAR database at the first exon of human *GPR1-AS* and mouse *Liz*, respectively (***Figure 5B and D***). Since *TFAP2C* and *ZSCAN4C* are activated in the placental lineage and preimplantation embryos, respectively (***Zhang et al., 2019***; ***Papuchova and Latos, 2022***; ***Figure 5—figure supplement 3***), binding of these transcription factors may play a role in the transcriptional activation of *GPR1-AS* or *Liz* during embryogenesis, and in turn imprinting of *ZDBF2*.

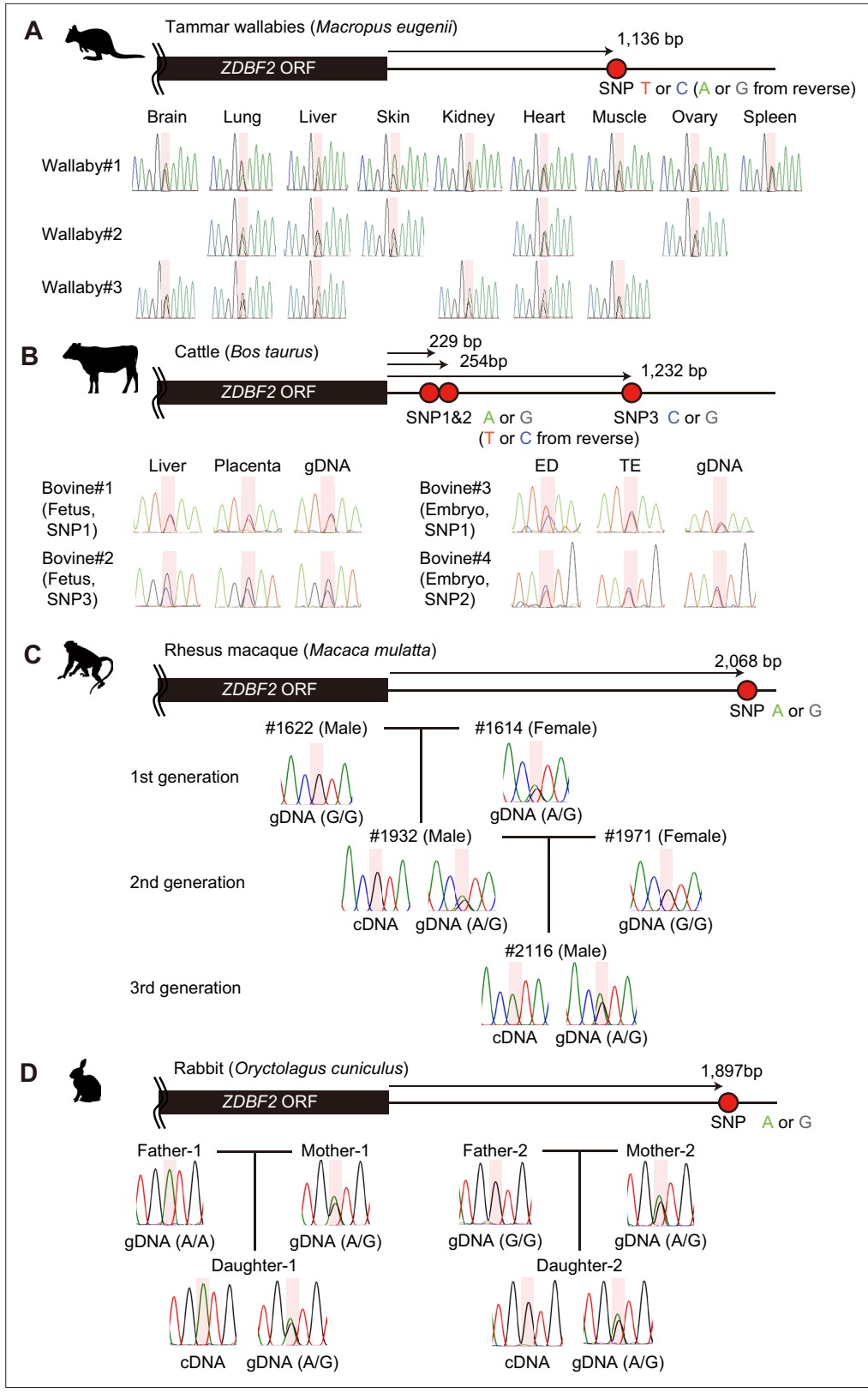

**Figure 3.** Allele-specific RT-PCR sequencing of *ZDBF2* in various mammals. Heterozygous genotypes were used to distinguish between parental alleles in adult tissues from tammar wallabies (**A**), fetal/embryonic tissues from cattle (**B**), blood samples from rhesus macaques (**C**), and rabbits (**D**), respectively. Primers were designed to amplify

*Figure 3 continued*
the 3'-UTR regions of *ZDBF2* orthologs and detect SNPs. Each SNP position is highlighted in red. Reverse primers were also used for Sanger sequencing. Animal silhouettes were obtained from PhyloPic.

The online version of this article includes the following figure supplement(s) for figure 3:

**Figure supplement 1.** Search for germline DMRs from oocyte and sperm DNA methylomes.

### *GPR1-AS* transcription and LTR reactivation in human

*GPR1-AS* is reported to be expressed only briefly during embryogenesis, both before and after implantation, with the exception of the placental lineage. For instance, human *GPR1-AS* has been identified in ES cells, and mouse *Liz* expression has been observed in blastocyst-to-gastrulation embryos but not in gametes (*Kobayashi et al., 2012b*; *Kobayashi et al., 2013*; *Duffié et al., 2014*). Therefore, it is likely that the expression of *GPR1-AS*/*Liz* is activated during preimplantation. To investigate the timing of *GPR1-AS* activation during embryonic development, we obtained public human RNA-seq data from the oocyte to blastocyst stages and performed expression analysis and transcript prediction (*Kai et al., 2022*; *Zou et al., 2022b*). Among the datasets from oocytes; zygotes; 2-, 4-, and 8 cell embryos; inner cell mass (ICM); and TE from blastocysts, *GPR1-AS* was only detected at the 8 cell stage, ICM, and TE; *GPR-AS-like* transcripts were detected as predicted transcripts, with TPM values above 1 (*Figure 6*). This indicates that *GPR1-AS* expression likely begins at the 8 cell stage in humans. However, at the *GPR1-ZDBF2* locus, both *GPR1* and *ZDBF2* were expressed in oocytes, but only *GPR1* was downregulated following fertilization and became undetectable beyond the 8 cell stage (*Figure 6* and *Figure 5—figure supplement 3*). We also examined MER21C expression at each stage and found that it increased from the 4-cell to 8-cell stage and peaked at the TE, coinciding with the appearance of *GPR1-AS* (*Figure 6—figure supplement 1*). Moreover, focusing on Kruppel-associated box zinc-finger proteins (KRAB-ZFPs) involved in the suppression of transposable elements, we observed that *ZNF789*—a key KRAB-ZFP that binds MER21C—was specifically downregulated at the 4 cell stage (*Figure 5—figure supplement 3*; *Imbeault et al., 2017*). Taken together, these results suggest that *GPR1-AS* expression begins in the preimplantation embryo, concurrently with a transcription factor/repressor milieu permissive for expression from its MER21C-derived promoter.

To determine whether specific epigenetic marks are associated with the LTR-derived promoter located in the first exon of human *GPR1-AS* and mouse *Liz*, we surveyed the ChIP-Atlas database (*Zou et al., 2022a*; *Figure 7—figure supplement 1*). Both sequences that constitute the oocyte-derived germline DMR exhibit an enrichment of H3K4me3 in sperm or male germ cells, consistent with DNA hypomethylation in sperm. These sequences also show an enrichment of H3K27ac in pluripotent stem cells (such as ES and iPS cells). Although, as discussed above, the first exon of mouse *Liz* has not been computationally determined as an LTR (according to the RepBase database), the mouse sequence is also enriched for H3K9me3, a repressive mark which is deposited at retrotransposons, in multiple cell types and tissues. Additionally, human and mouse germline DMRs coincided with the TFAP2C peak (trophoblast stem cells) and ZSCAN4C peak (ES cells), respectively, and aligned with the binding motif predictions found in the JASPAR database (*Figure 5B and D*). These data indicate that the first exons of human *GPR1-AS* and mouse *Liz* exhibit similar epigenomic modification variations from gamete to embryonic development, with their promoter activities induced after fertilization. However, the specific regulatory pathways governing this activation may differ between species, perhaps reflected in differences in the specific set of transcription factors involved (*Figure 7*).

## Discussion

The *ZDBF2* gene, the last canonical imprinting gene identified in mice and humans, was recently shown to regulate neonatal feeding behavior (*Glaser et al., 2022*). This discovery has garnered attention owing to its implications for behavioral phenotypes, aligning with the conflicting hypothesis of genomic imprinting mechanisms. The conflict hypothesis proposes that the evolutionary significance of genomic imprinting mechanisms lies in the functional conflicts between paternally and maternally expressed genes, resulting from the conflicting genomic survival strategies of the father, child, and mother (*Moore and Haig, 1991*). *ZDBF2* does not directly influence cell differentiation or ontogeny,

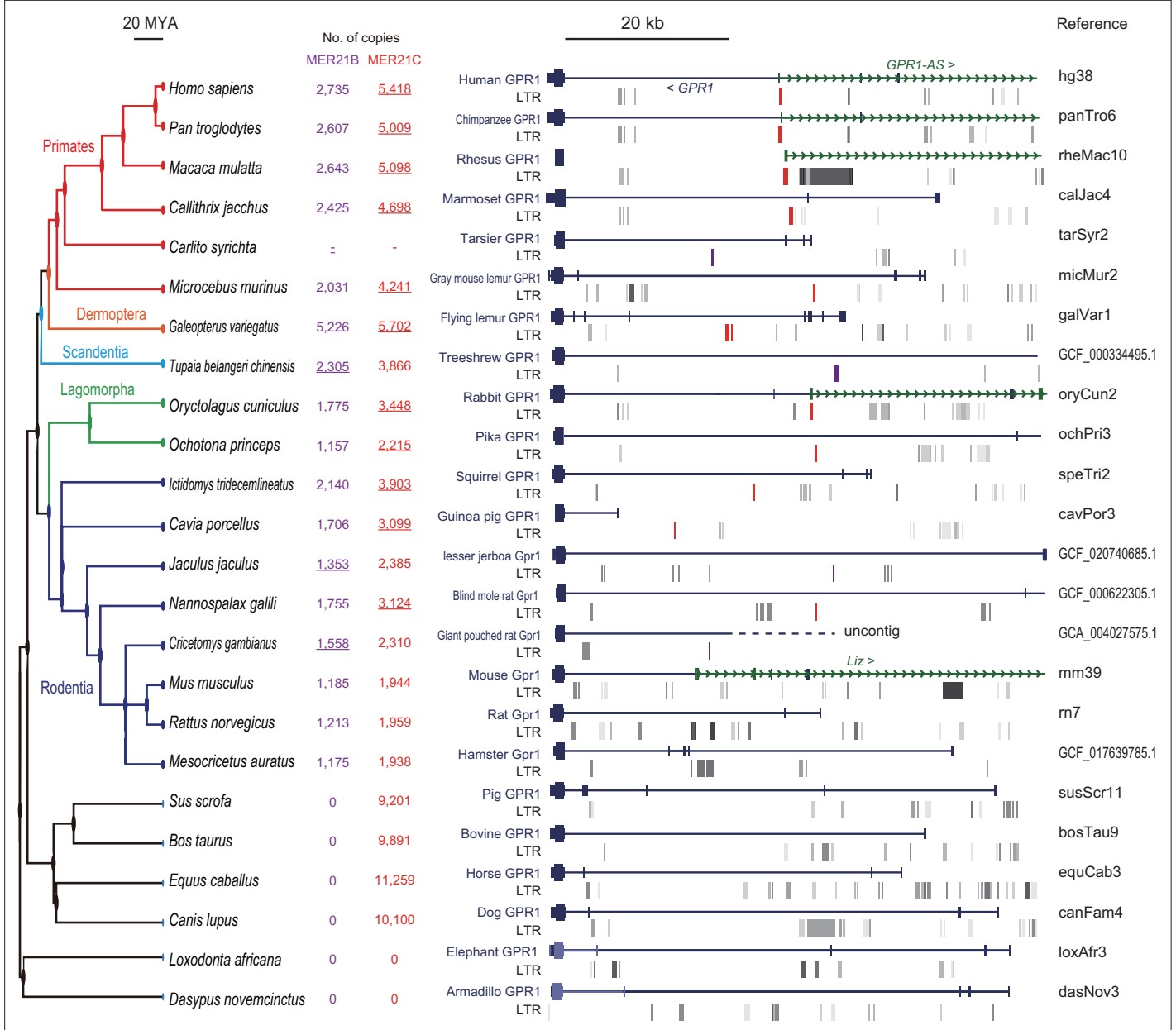

**Figure 4.** Multi-species comparison of long terminal repeat (LTR) retrotransposon locations at *GPR1* locus. A total of 24 mammalian genomes were compared, including six primates (human, chimpanzee, rhesus macaque, marmoset, tarsier, and gray mouse lemur), one colugo (flying lemur), one treeshrew (Chinese treeshrew), two lagomorphs (rabbit and pika), eight rodents (squirrel, guinea pig, lesser jerboa, blind mole rat, giant pouched rat, mouse, rat, and golden hamster), and six other eutherians (pig, cow, horse, dog, elephant, and armadillo). Among the selected genomes, LTRs that can be considered homologous to MER21C, which corresponds to the first exon of *GPR1-AS*, are marked in red. In tarsier, treeshrew, lesser jerboa, and giant pouched rat, the orthologous LTRs were annotated as MER21B, which exhibits 88% similarity with MER21C in their consensus sequences through pairwise alignment. MER21B is marked in purple. According to Dfam, the MER21C and MER21B subfamilies are specific to the genomes of Boroeutherians and Euarchontoglires, respectively. The copy number of MER21C/B in selected species is shown in red and purple (LTRs likely matching the *GPR1-AS* exon are underlined). There are 5418 and 2529 copies of MER21C and 2894 and 1535 copies of MER21B in human and mouse genomes, respectively.

The online version of this article includes the following figure supplement(s) for figure 4:

**Figure supplement 1.** Reanalysis of repeat positions using RepeatMasker.

**Figure supplement 2.** Multiple genome alignments at the first exon of *GPR1-AS* locus.

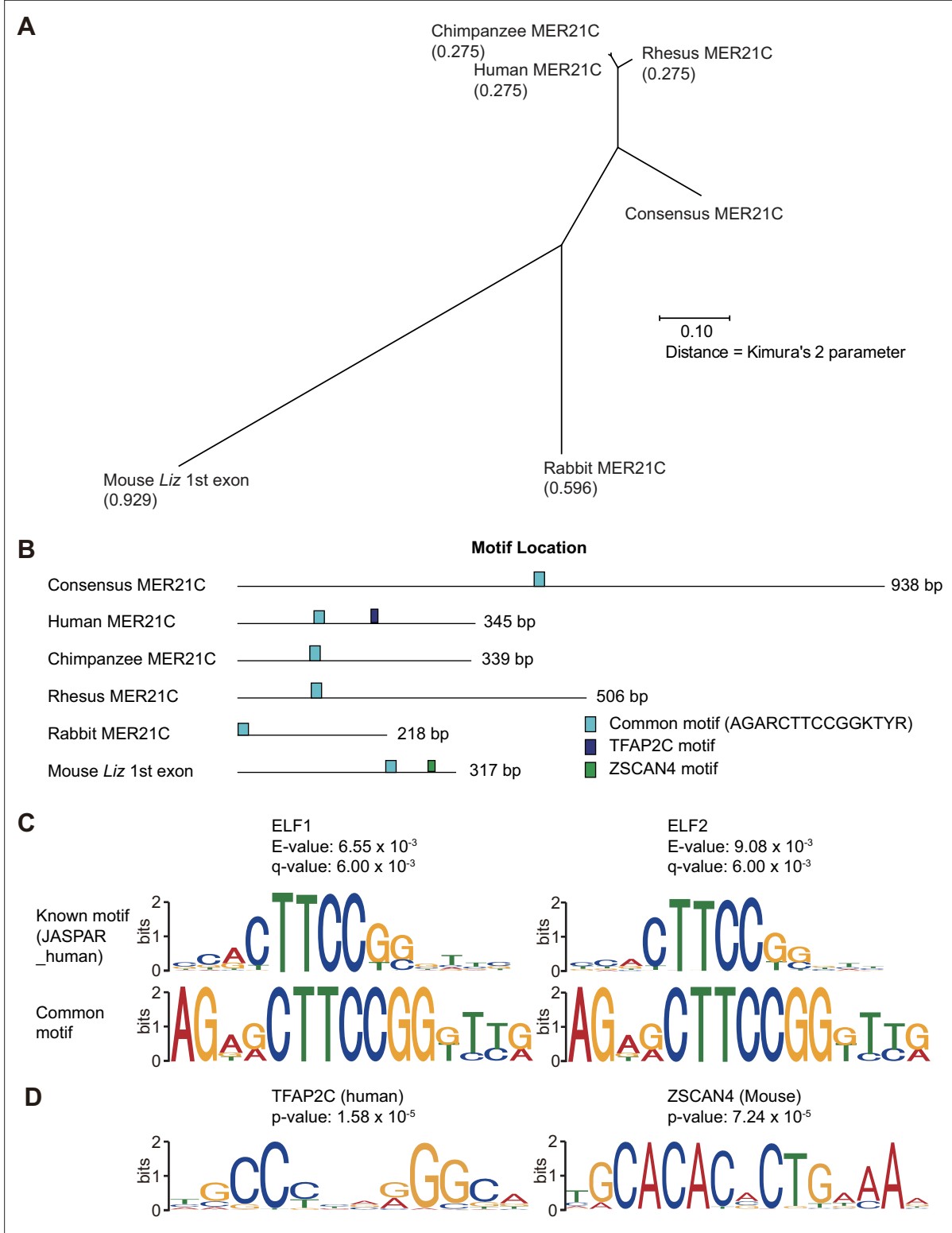

**Figure 5.** Comparison of MER21C-derived sequences overlapping the first exon of *GPR1-AS* orthologs. (**A**) Phylogenetic tree of MER21C-derived sequences estimated by multiple sequence alignment (MSA) using multiple sequence comparison by log-expectation (MUSCLE) program. (**B**) Positions of common and unique cis-acting elements at each sequence. (**C**) Motif structures of the common region that contains E74-like factor 1 and 2 (ELF1 and ELF2) binding motifs. (**D**) Motif structures of transcription factor AP-2 gamma (TFAP2C) and Zinc finger and SCAN domain containing 4 (ZSCAN4).

The online version of this article includes the following figure supplement(s) for figure 5:

*Figure 5 continued on next page*

*Figure 5 continued*

**Figure supplement 1.** Pairwise alignment between consensus sequences of retrotransposons and *GPR1-AS*-exonic MER21 sequences.

**Figure supplement 2.** Promoter activities of first exons of mouse *Liz* and human *GPR1-AS*.

**Figure supplement 3.** Expression patterns of transcription factors and imprinted genes during human preimplantation development.

such as embryogenesis, but rather affects individual growth through an indirect pathway involving feeding behavior. In humans, 'imprinting diseases,' such as Prader-Willi syndrome and Angelman syndrome, cause psychiatric disorders and growth retardation (***Nicholls, 2000***). Considering that many imprinted genes show specialized expression patterns in the brain, *ZDBF2* is particularly interesting as a factor that can induce behavioral effects. However, the evolutionary origin of *ZDBF2* imprinting had remained unclear. In this study, we report that *ZDBF2* does not exhibit imprinted expression in cattle and tammar wallabies, mammalian species that do not belong to the Euarchontoglires clade. In contrast, we identified *GPR1-AS* orthologs in all the Euarchontoglires species examined, including

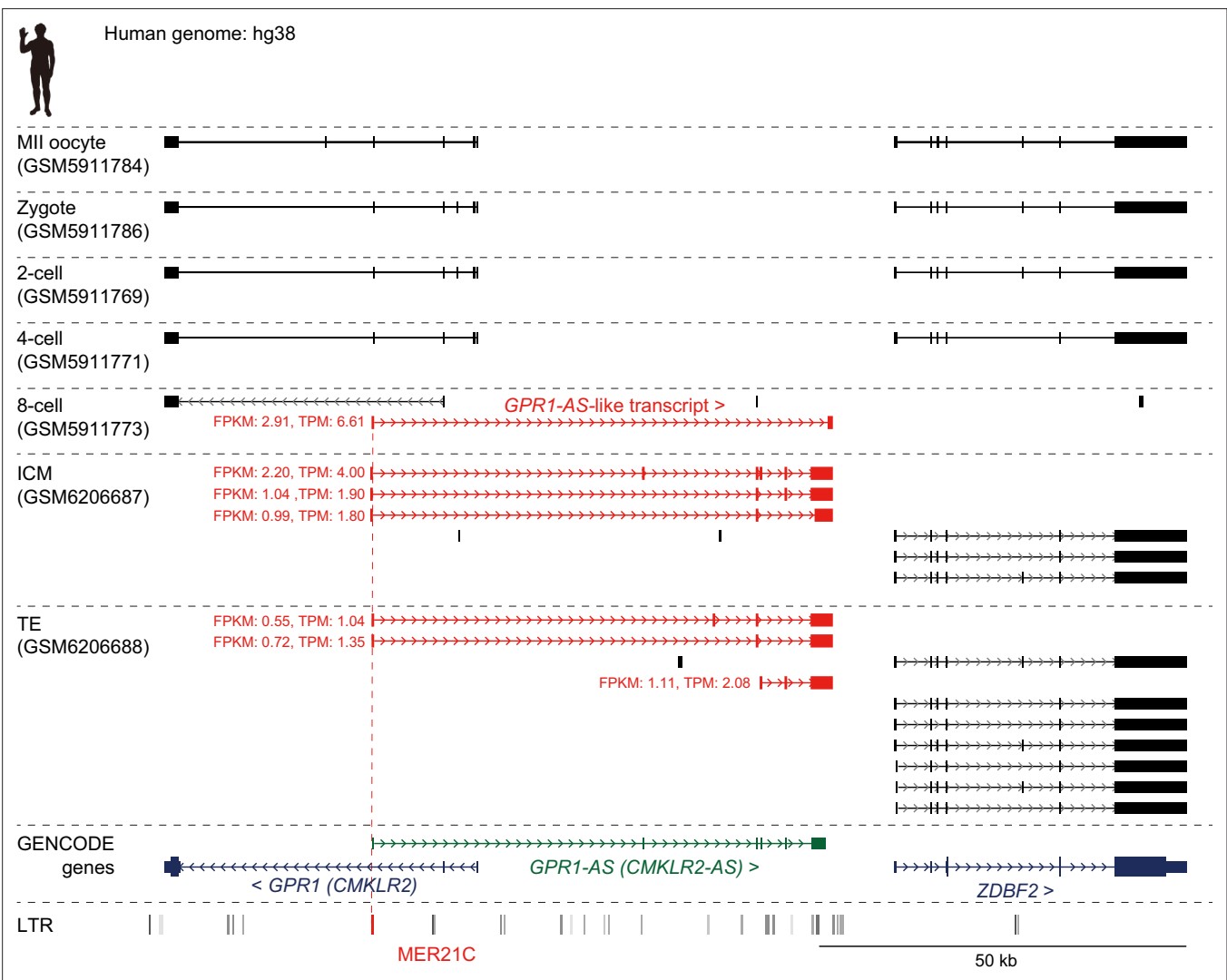

**Figure 6.** Initiation of *GPR1-AS* transcription before implantation. Genome browser screenshots of the *GPR1-ZDBF2* locus in humans at preimplantation stages, including the MII oocyte, zygote, 2 cell, 4 cell, 8 cell, inner cell mass (ICM), and trophectoderm (TE) from the blastocyst. Predicted transcripts were generated from publicly available full-length RNA-seq datasets, with detected *GPR1-AS*-like transcripts and their fragments per kilobase million (FPKM) and transcripts per kilobase million (TPM) values highlighted in red. Silhouette was obtained from PhyloPic.

The online version of this article includes the following figure supplement(s) for figure 6:

**Figure supplement 1.** Human long terminal repeat (LTR) reactivation during preimplantation development.

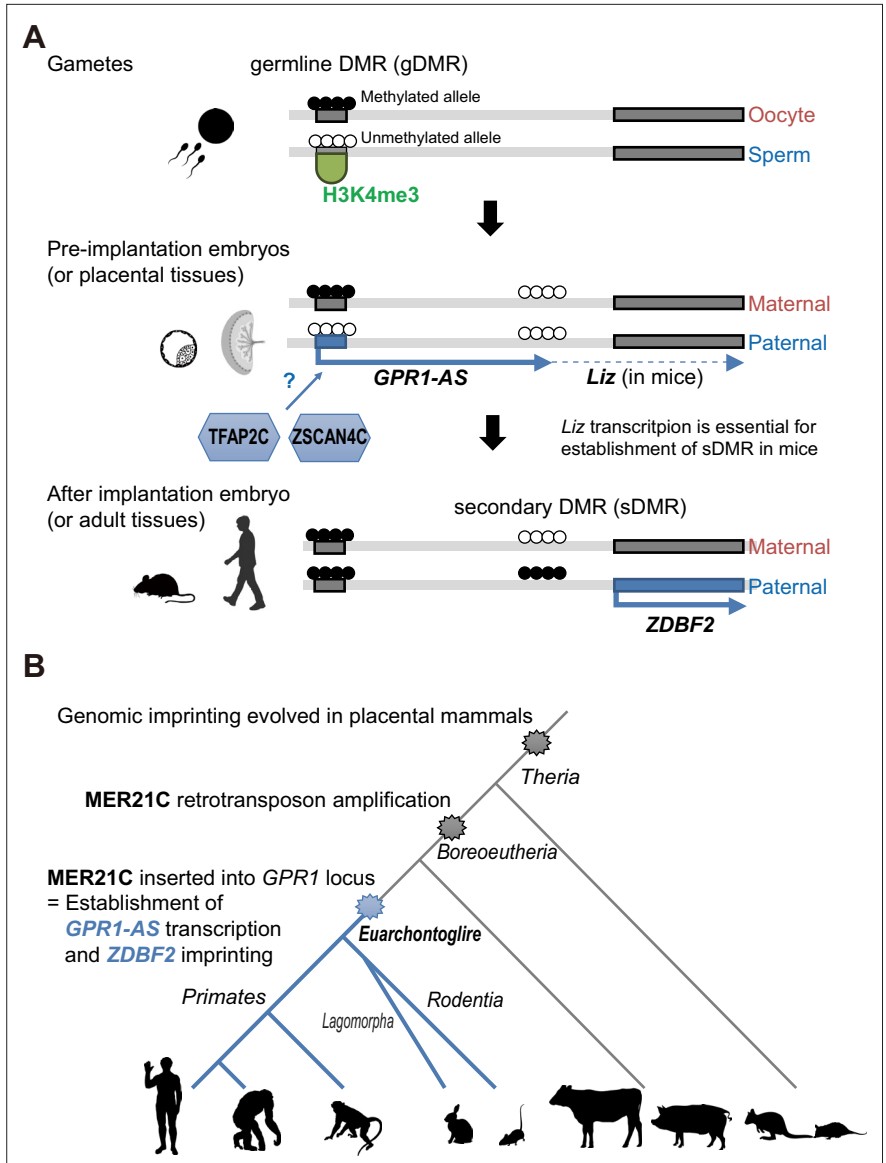

**Figure 7.** Establishment of *ZDBF2* imprinted domain in evolution and genome biology. (**A**) Scheme of epigenetic and transcriptional changes at the first exon of mouse *Liz* and human *GPR1-AS*. (**B**) Timescale of the evolution of *ZDBF2* imprinting and LTR (MER21C) insertion. Animal silhouettes were obtained from PhyloPic (mouse silhouette by Katy Lawler, available under a CC BY 4.0 license; opossum silhouette by Sarah Werning, available under a CC BY 3.0 license).

The online version of this article includes the following figure supplement(s) for figure 7:

**Figure supplement 1.** Interspecies epigenomic comparisons between human *GPR1-AS* and mouse *Liz*.

humans, chimpanzees, baboons, rhesus macaques, and rabbits. We propose that the first exon of *GPR1-AS* is derived from a MER21C retrotransposon and that the insertion of this LTR at the *GPR1* intragenic region in the common ancestor of Euarchontoglires was the crucial event in the genesis of imprinting of the *ZDBF2* gene (***Figure 7***).

Two paternally expressed imprinted genes, *PEG10/SIRH1*, and *PEG11/RTL1/SIRH2*, that encode GAG-POL proteins of *sushi-ichi* LTR retrotransposons have been identified in mammals and are essential for placenta formation and maintenance (***Kaneko-Ishino and Ishino, 2012***). The presence of these genes provides evidence that LTR insertions have played a significant role in genomic imprinting as well as placentation throughout evolution of placental mammals. In contrast, while the *ZDBF2* gene is

present in all vertebrate genomes, it is only imprinted in mammals belonging to the Euarchontoglires superorder of placental mammals. We previously demonstrated that transcriptional activity of specific LTRs in oocytes results in species-specific imprinting of numerous genes in rodents and primates (*Bogutz et al., 2019*). These LTRs act as alternative promoters and produce unique transcripts in the oocyte, which we referred to as LITs (LTR-initiated transcripts). Such transcription leads to a high level of methylation in the transcribed region, similar to gene body methylation, and is responsible for extensive interspecies differences in DNA methylation (*Brind'Amour et al., 2018*). The key difference between these LTRs and the LTR retrotransposon focused on here is the stage in development when LTR (re)activation occurs, i.e., before or after fertilization, respectively. Some LITs, which are driven by LTR families that are particularly active in oocytes, become active during oogenesis and establish oocyte-derived gDMRs of species-specific imprinted genes, such as *Impact* and *Slc38a4* genes in mice. Strikingly, deletion of these LTRs causes loss of imprinting in mouse (rodent)-specific imprinted genes (*Bogutz et al., 2019*). *GPR1-AS* is a (likely noncoding) RNA whose transcription is initiated from an LTR-derived sequence, similar to the LITs found in oocytes; however, transcription in this case initiates after fertilization. In mice, *Liz*, a long isoform of *Zdbf2* and putative *Gpr1-as* ortholog, is essential for establishing a somatic secondary DMR upstream of *Zdbf2* (*Greenberg et al., 2017*). Although the first exon of *Liz* does not overlap with an annotated LTR, MER21C (or MER21B, a closely related LTR family) insertions are found in the homologous region in most rodents, with the exception of mice, rats, and hamsters. Our analyses suggest that a MER21C relic is also present in these rodents, but has accumulated a number of mutations and in turn is no longer recognized as a MER21C element using optimal pairwise alignment algorithms such as RepeatMasker. Indeed, multiple genome alignment using Cactus reveals that the first exon of human *GPR1-AS* is homologous across all selected rodents, including mouse, rat, and hamster. Based on these observations, we propose that *Liz* and *GPR1-AS* are bona fide LITs with a MER21C-derived sequence serving as an alternative promoter. A combined approach involving the annotation of repetitive elements with tools like RepeatMasker and the reconstruction of ancestral genomes using multiple-genome alignments can uncover highly degenerated LTR relics. For instance, Didier Trono's group recently identified numerous previously unannotated transposable elements in the human genome using a similar methodology (*Matsushima et al., 2024*). Advances in such computational analyses could enable the identification of additional functional LTR relics and provide insights into the origin of lineage-specific regulatory sequences that are embedded within newly discovered lineage-specific LTR relics.

Transposable elements are recognized by DNA-binding factors and their co-repressors and are suppressed through epigenetic and post-transcriptional regulation in both germline and somatic tissues to protect host genome stability. For example, LTR-retrotransposons are bound by KRAB-ZFPs, which recruit the KAP1/SETDB1 co-repressor complex, promoting deposition of the repressive histone modification H3K9me3 (*Matsui et al., 2010*; *Karimi et al., 2011*). However, numerous retrotransposons are activated and robustly expressed during maternal to zygotic transition, where dynamic epigenetic reprogramming occurs. Indeed, a subset of transposable elements have been co-opted by the host during this window of development, including in support of embryonic development (*Sakashita et al., 2023*). In humans, retrotransposons are activated at the 8 cell stage and gradually downregulated in later developmental stages, coinciding with embryonic genome activation. In addition to the full-length endogenous retrovirus (ERV) sequences with partial coding potential, approximately 85% of human ERVs (HERVs) exist as solitary LTRs, known as 'solo LTRs,' which provide a rich source of cis-regulatory elements for gene expression during human embryonic development. Retrotransposons located near host genes can act as alternative promoters or regulatory modules, such as enhancers, to activate embryonic genes (*Grow et al., 2015*; *Hashimoto et al., 2021*). The discovery of the *GPR1-AS* as a transcript that initiates in an ancestral LTR reveals a new aspect of the functional role of these ectopic regulatory elements, with activation in this case occurring after fertilization.

In summary, this study reveals that the origin of imprinting of *ZDBF2* can be traced back to a common ancestor of Euarchontoglires, in which an LTR element inserted in the locus. The solo LTR derived from this element (following recombination between 5' and 3' LTRs) drives expression of *GPR1-AS*, which in turn plays a critical role in establishing imprinting of the *ZDBF2* locus, including in both mice and humans. Although the house mouse is used widely in the laboratory, and the *Mus musculus* genome has been extensively characterized, the retroviral origin of this promoter would

not have been identified if this analysis had been performed solely on mice, as the annotated exon 1 of *Liz* (alternatively named *Gpr1-as, Platr12*, or *Zdbf2linc*) is highly degenerate and in turn not recognized as an LTR in this species. Comparative analysis of multi-omics data, including genomes, transcriptomes, and epigenomes, across species was critical for the identification of the central role of a MER21C LTR in imprinting of *ZDBF2*. This study adds to a growing list of LTR elements that have been domesticated in mammals, with their transcriptional activity serving as a mechanism for the genesis of imprinting, including for a number of genes during gametogenesis and in the case of *ZDBF2*, following fertilization.

## Methods

### Ethical approval for animal work

All animal experiments undertaken in Japan were approved by the Institutional Animal Care and Use Committees of the following institutions and were conducted in accordance with the *Guidelines for Proper Conduct of Animal Experiments* by the Science Council of Japan (2006): the Center for the Evolutionary Origins of Human Behavior of Kyoto University (Approval No. 2018–004 for chimpanzee and No. 2023–134 for rhesus macaque), Hokkaido University (Approval No. 19-0162 for cow), Meiji University (Approval No. IACUC17-0005 for pig), RIKEN Kobe Branch (Approval No. A2001-03 for opossum), and Kitayama Labes (Approval Nos. IBC56-028 and MINOWA-61–011 for rabbit). Experiments involving the tammar wallaby were approved by the University of Melbourne Animal Experimentation and Ethics Committees (Ethics ID: 1413134.1) and conducted in accordance with the *Australian code for the care and use of animals for scientific purposes* (**National Health and Medical Research Council, 2013**).

### Animals for transcriptome analysis

Placental samples from chimpanzees (*Pan troglodytes verus*) were provided by Kumamoto Zoo and Kumamoto Sanctuary via the Great Ape Information Network. The samples were stored at −80°C before use. Total RNA was isolated from the placenta using an AllPrep DNA/RNA Mini Kit (Qiagen, Netherlands). Bovine (*Bos taurus*, Holstein cow) embryos were prepared by in vitro oocyte maturation, fertilization (IVF), and subsequent in vitro embryo culture (**Akizawa et al., 2018**). Briefly, presumptive IVF zygotes were denuded by pipetting after 12 hr of incubation and cultured up to the blastocyst stage in mSOFai medium at 38.5°C in a humidified atmosphere of 5% $CO_2$ and 5% $O_2$ in air for 8 d (D8). D8 blastocysts were further cultured on agarose gel for 4 d as described previously (**Akizawa et al., 2018**; **Saito et al., 2022**). The embryo proper (embryonic disc: ED) and TE portions of IVF D12 embryos were mechanically divided using a microfeather blade (Feather Safety Razor, Japan) under a stereomicroscope. Total RNA from ED and TE samples was isolated using the ReliaPrepTM RNA Cell Miniprep System (Promega, MA, USA) according to the manufacturer's instructions. Rabbit embryos were prepared on embryonic day 6.75 (E6.75) by mating female Dutch and male JW rabbits (Kitayama Labes, Japan). Pig embryos (*Sus scrofa domesticus*: Landrace pig) at E15 and opossum embryos at E11 were prepared (**Kiyonari et al., 2021**). Individual embryo proper and TE were mechanically divided using a 27-30G needle and tweezer under a stereomicroscope with 10% FCS/M199 -medium in rabbits, HBSS medium in pigs, and 3% FCS/PBS in opossums. Total RNA was isolated from individual tissues using the RNeasy Mini and Micro Kit (Qiagen). The quality of the total RNA samples was assessed using an Agilent 2100 Bioanalyzer system (Thermo Fisher Scientific, MA, USA) and high-quality RNA samples (RIN≧7) were selected for RNA-seq library construction.

### Strand-specific RNA library preparation and sequencing

Embryonic total RNA (10 ng) from opossums and pigs, 5 ng of placental total RNA from chimpanzees, and 1 ng of embryonic total RNA from rabbits and cattle were reverse transcribed using the SMARTer Stranded Total RNA-Seq Kit v2 - Pico Input Mammalian (Takara Bio, Japan) according to the manufacturer's protocols, where Read 2 corresponds to the sense strand due to template-switching reactions. RNA-seq libraries were quantified by qPCR using the KAPA Library Quantification Kit (Nippon Genetics, Japan). All libraries were pooled and subjected to paired-end 75 bp sequencing (paired-end 76 nt reads with the first 1 nt of Read 1 and the last 1 nt of Read 2 trimmed) using the NextSeq500

system (Illumina, CA, USA). For each library, the reads were trimmed using Trimmomatic to remove two nucleotides from the 5' end of the Read 2.

## RNA-seq data set download

Strand-specific RNA-seq datasets from the human bulk placenta, rhesus macaque trophoblast stem cells, and mouse bulk placenta at E16.5, were downloaded from the accession numbers SRR12363247 and SRR12363248 for humans, SRR1236168 and SRR1236169 for rhesus macaques, and SRR943345 for mice (*Necsulea et al., 2014*; *Rosenkrantz et al., 2021*). Non-directional RNA-seq data from placentas or extra-embryonic tissues of 15 mammalian species, including humans, bonobos, baboons, mice, golden hamsters, rabbits, pigs, cattle, sheep, horses, dogs, bats, elephants, armadillos, and opossums, were downloaded (*Armstrong et al., 2017*; *Mika et al., 2022*). Full-length RNA-seq (based on the smart-seq method) data from human oocytes, zygotes, 2-, 4-, and 8-cell, ICM, and TE were downloaded from previous studies (*Kai et al., 2022*; *Zou et al., 2022b*). The corresponding accession numbers are shown in *Supplementary file 2*.

## RNA-seq data processing

All FASTQ files were quality-filtered using Trimmomatic and mapped to individual genome references (*Supplementary file 2*) using Hisat2 with default parameters and StringTie2 with default parameters and '-g 500' option. Fragments per kilobase million (FPKM) and transcripts per kilobase million (TPM) were calculated using StringTie2 with '-e' option and the GENCODE GTF file, selecting lines containing the 'Ensembl_canonical' tag. Read counts of human transposable elements were computed at the subfamily level using the TEcount. The read counts were normalized by the total number of mapped reads to retrotransposons in each sample as Reads per kilobase million (RPKM).

## Data visualization

All GTF files (StringTie2 assembled transcripts) were uploaded to the UCSC Genome Browser, and the predicted transcripts, annotated genes (GENCODE, Ensembl, or RefSeq), and LTR locations were visualized. The expression profiles of human *GPR1*, *GPR1-AS*, and *ZDBF2* in the somatic tissues (*Fagerberg et al., 2014*; *Duff et al., 2015*), and these imprinted genes, the associated transcription factors (*TFAP2C*, *ZSCAN4*, *ELF1*, and *ELF2*), and LTR subfamilies in early embryo (*Kai et al., 2022*; *Zou et al., 2022b*) were downloaded or calculated, and visualized as heatmaps using the Morpheus software. The estimated evolutionary distance between the selected mammals was visualized as a physiological tree with a branch length of millions of years (MYA) using Timetree. The pairwise alignment, multiple sequence alignment, and physiological tree of MER21C sequences overlapping *GPR1-AS* were conducted with Genetyx software (Genetyx, Japan) using the MSA and MUSCLE programs. The common cis-motif was identified and compared with known transcription factor binding motifs from JASPAR CORE database using the XSTREAM and TOMTOM programs in the MEME suite. JASPAR motifs, logos, and *p*-values were obtained from the JASPAR hub tracks of the UCSC Genome Browser and JASPAR database. LTR positions of human (hg19) and mouse (mm10) were downloaded from UCSC Genome Browser and re-uploaded to Integrative Genomics Viewer with epigenetic patterns from ChIP-Atlas (*Zou et al., 2022a*) and oocyte/sperm DNA methylomes (*Brind'Amour et al., 2018*).

## Allele-specific expression analysis

Tammar wallabies (*Macropus eugenii*) of Kangaroo Island origin were maintained in our breeding colony in grassy, outdoor enclosures. Lucerne cubes, grass, and water were provided ad libitum and supplemented with fresh vegetables. Pouch young were dissected to obtain a range of tissues. DNA/RNA was extracted from multiple tissues of the Tammar wallaby using TRIzol Reagent (Thermo Fisher Scientific), and cDNA synthesis was performed using Transcriptor First Strand cDNA Synthesis Kit (Roche, Switzerland). PCR was performed using TaKaRa Ex Taq Hot Start Version (TaKaRa Bio) with primers designed at 3'UTR of wallaby *ZDBF2* according to the manufacturer's instructions. Fetal tissues of pregnant healthy Holstein cattle were obtained from a local slaughterhouse. DNA/RNA was extracted from bovine fetus (liver), placentas, and in vitro cultured embryos using ISOGEN (Wako, Japan), and cDNA synthesis and PCR were performed using SuperScript IV One-Step RT-PCR System (Thermo Fisher Scientific) with primers designed at 3'UTR of bovine *ZDBF2* according to the manufacturer's instructions. Target regions were also amplified using genomic DNA and Takara EX Taq Hot

Start Version (Takara Bio). Whole blood samples of rhesus macaque (*Macaca mulatta*) were obtained from the Primate Research Institute (PRI) of Kyoto University, and those of Japanese White rabbit (*Oryctolagus cuniculus*) were purchased from Kitayama Labes through Oriental Yeast Co., Ltd. (Japan). DNA/RNA was extracted from blood samples using ISOSPIN Blood & Plasma DNA (Nippon Gene, Japan) and Nucleospin RNA Blood (Takara Bio) and cDNA synthesis and PCR were performed using SuperScript IV One-Step RT-PCR System (Thermo Fisher Scientific) with primers designed at 3'UTR of rhesus macaque and rabbit *ZDBF2* according to the manufacturer's instructions. Sanger sequencing was performed using a conventional Sanger sequencing service (Fasmac, Japan) and ABI 3130xl Genetic Analyzer (Thermo Fisher Scientific). Sanger sequencing results were visualized using 4Peaks. Primers used in this analysis are listed in *Supplementary file 3*.

### DNA methylome analysis

The DSS R package (v2.54.0) was used to identify DMRs from oocyte and sperm CpG report files (accession numbers GSM1466810, GSM1466811, and GSE143849), with smoothing enabled and customized parameters. These parameters included a minimum CpG methylation difference of 50%, a p-value threshold of 0.05, a minimum DMR length of 200 bp, and other default settings. All BED files were uploaded to the UCSC Genome Browser and CpG sites with DNAme levels and called DMR positions were visualized.

### Dual reporter assay

Luciferase reporter plasmids were constructed using the pGL4.13 vector (Promega). Sequences containing the first exon and upstream regions of both mouse *Liz* and human *GPR1-AS* were amplified by PCR using TaKaRa HS Perfect Mix (Takara Bio) and specific primers with NheI and HindIII recognition sites at their 5' ends (*Supplementary file 3*). The PCR products were inserted between the NheI and HindIII sites of the pGL4.13 vector using DNA Ligation Mix (Takara Bio). To evaluate the promoter activity of these sequences, the SpectraMax DuoLuc Reporter Assay Kit (Molecular Devices, CA, USA) was employed. Human embryonic kidney 293T (HEK293T) cells were cultured in Dulbecco's Modified Eagle's Medium (Thermo Fisher Scientific, Cat. No. 10313021) supplemented with 10% fetal bovine serum (NICHIREI Biosciences, Tokyo, Japan), 0.1 mg/mL penicillin-streptomycin (Thermo Fisher Scientific), and GlutaMax (Thermo Fisher Scientific, Cat. No. 35050061), on gelatin-coated dishes. For luciferase reporter assays, HEK293T cells were seeded in gelatin-coated 96-well plates at a density of $4 \times 10^5$ cells/well prior to transfection. Transient transfections were carried out using Lipofectamine 2000 Transfection Reagent (Thermo Fisher Scientific) according to the manufacturer's instructions. Each well was co-transfected with 10 ng of the pGL4.75 plasmid (Promega) as an internal transfection control reporter and 100 ng of the pGL4.13 plasmid with or without the candidate regulatory elements. After 48 hr of transfection, dual-luciferase assays were performed as per the manufacturer's protocol (Molecular Devices). Luminescence signals were measured using a SpectraMax iD3 Multi-Mode Microplate Reader (Molecular Devices). Firefly luciferase activities (pGL4.13) were normalized to Renilla luciferase activities (PGL4.75).

### Software used

Trimmomatic (http://www.usadellab.org/cms/?page=trimmomatic; *Bolger et al., 2014*; RRID:SCR_011848)

Hisat2 (http://daehwankimlab.github.io/hisat2/; *Kim et al., 2019*; RRID:SCR_015530)

StringTie2 (https://ccb.jhu.edu/software/stringtie/; *Kovaka et al., 2019*; RRID:SCR_011848)

TEcount (https://github.com/bodegalab/tecount/; *Jin et al., 2015*; *Polimeni, 2023*)

Integrative Genomics Viewer (IGV: https://software.broadinstitute.org/software/igv/; *Robinson et al., 2011*; RRID:SCR_016323)

4Peaks (https://nucleobytes.com/4peaks/; *Nucleobytes, 2025*)

DSS (https://bioconductor.org/packages/release/bioc/html/DSS.html; *Feng et al., 2014*; RRID:SCR_002119)

### Web-tools used

UCSC Genome Browser (https://genome.ucsc.edu/)

Morpheus (https://software.broadinstitute.org/morpheus/)
Timetree (http://timetree.org/)
MEME Suite (https://meme-suite.org/meme/)
JASPAR (https://jaspar.genereg.net/)
ChIP-Atlas (https://chip-atlas.org/)
PhyloPic (https://www.phylopic.org/)
RepeatMasker (https://www.repeatmasker.org/)
PipMaker and MultiPipMaker (http://pipmaker.bx.psu.edu/pipmaker/)

## Acknowledgements

HK was supported by MEXT KAKENHI Grant Numbers JP21H02382 and JP25K02190, and KK was supported by JP20H00471. HK and HI were supported by The Cooperative Research Programs of the NODAI Genome Research Center at Tokyo University of Agriculture, and the Center for the Evolutionary Origins of Human Behavior and Wildlife Research Center at Kyoto University. HK received additional support from the Mitsubishi Foundation and the Takeda Science Foundation. Additionally, we acknowledge Mr. Yasuyuki Osada (Kitayama Laboratories, Japan) for his technical assistance in handling the rabbit embryos, and Mr. Aaron Bogutz (University of British Columbia) for his helpful comments.

## Additional information

### Funding

| Funder | Grant reference number | Author |
|---|---|---|
| Japan Society for the Promotion of Science | Grant-in-Aid for Scientific Research (B) JP21H02382 | Hisato Kobayashi |
| Japan Society for the Promotion of Science | Grant-in-Aid for Scientific Research (B) JP23K21282 | Hisato Kobayashi |
| Japan Society for the Promotion of Science | Grant-in-Aid for Scientific Research (B) JP25K02190 | Hisato Kobayashi |
| Japan Society for the Promotion of Science | Grant-in-Aid for Scientific Research (A) JP20H00471 | Kazuki Kurimoto |
| The Cooperative Research Program of the NODAI Genome Research Center at Tokyo University of Agriculture | | Hisato Kobayashi |
| The Cooperative Research Program of the Center for the Evolutionary Origins of Human Behavior and Wildlife Research Center at Kyoto University | | Hiroo Imai |
| Mitsubishi Foundation | Grant No. 202310022 | Hisato Kobayashi |
| Takeda Science Foundation | 2023 Medical Research Grant Application No. 2023040490 | Hisato Kobayashi |
| Canadian Institutes of Health Research | PJT-190055 | Matthew Lorincz |

The funders had no role in study design, data collection and interpretation, or the decision to submit the work for publication.

## Author contributions

Hisato Kobayashi, Conceptualization, Data curation, Formal analysis, Supervision, Funding acquisition, Validation, Investigation, Visualization, Writing – original draft, Project administration, Writing – review and editing; Tatsushi Igaki, Data curation, Validation, Investigation; Soichiro Kumamoto, Keisuke Tanaka, Tomoya Takashima, Data curation; So I Nagaoka, Formal analysis, Methodology; Shunsuke Suzuki, Masaaki Hayashi, Marilyn B Renfree, Resources, Formal analysis; Manabu Kawahara, Shun Saito, Toshihiro Kobayashi, Hiroshi Nagashima, Hitomi Matsunari, Kazuaki Nakano, Ayuko Uchikura, Hiroshi Kiyonari, Mari Kaneko, Hiroo Imai, Kazuhiko Nakabayashi, Resources; Matthew Lorincz, Conceptualization, Writing – review and editing; Kazuki Kurimoto, Funding acquisition, Writing – review and editing

## Author ORCIDs

Hisato Kobayashi ⓘ https://orcid.org/0000-0003-3800-4691
Soichiro Kumamoto ⓘ https://orcid.org/0000-0002-3492-117X
Keisuke Tanaka ⓘ https://orcid.org/0000-0003-0658-7781
Tomoya Takashima ⓘ https://orcid.org/0000-0002-3648-1569
So I Nagaoka ⓘ https://orcid.org/0000-0002-2377-4936
Marilyn B Renfree ⓘ https://orcid.org/0000-0002-4589-0436
Manabu Kawahara ⓘ https://orcid.org/0000-0002-7172-6082
Toshihiro Kobayashi ⓘ https://orcid.org/0000-0001-8019-0008
Kazuaki Nakano ⓘ https://orcid.org/0000-0003-1306-6247
Ayuko Uchikura ⓘ https://orcid.org/0000-0002-1535-3808
Hiroshi Kiyonari ⓘ https://orcid.org/0000-0002-1509-8747
Hiroo Imai ⓘ https://orcid.org/0000-0003-0729-0322
Kazuhiko Nakabayashi ⓘ https://orcid.org/0000-0003-2927-0963
Matthew Lorincz ⓘ https://orcid.org/0000-0003-0885-0467
Kazuki Kurimoto ⓘ https://orcid.org/0000-0003-0206-414X

## Ethics

All animal experiments undertaken in Japan were approved by the Institutional Animal Care and Use Committees of the following institutions and were conducted in accordance with the Guidelines for Proper Conduct of Animal Experiments by the Science Council of Japan (2006): the Center for the Evolutionary Origins of Human Behavior of Kyoto University (Approval No. 2018-004 for chimpanzee and No. 2023-134 for rhesus macaque), Hokkaido University (Approval No. 19-0162 for cow), Meiji University (Approval No. IACUC17-0005 for pig), RIKEN Kobe Branch (Approval No. A2001-03 for opossum) and Kitayama Labes (Approval Nos. IBC56-028 and MINOWA-61-011 for rabbit). Experiments involving the tammar wallaby were approved by the University of Melbourne Animal Experimentation and Ethics Committees (Ethics ID: 1413134.1) and conducted in accordance with the Australian code for the care and use of animals for scientific purposes (NHMRC, 2013).

Reviewer #1 (Public review): https://doi.org/10.7554/eLife.94502.3.sa1
Reviewer #2 (Public review): https://doi.org/10.7554/eLife.94502.3.sa2
Reviewer #3 (Public review): https://doi.org/10.7554/eLife.94502.3.sa3
Author response https://doi.org/10.7554/eLife.94502.3.sa4

---

# Additional files

## Supplementary files

Supplementary file 1. Imprinted expression patterns of GPR1-AS and ZDBF2 among mammalian species.

Supplementary file 2. Lists of RNA-seq datasets used in this study.

Supplementary file 3. Primers used in this study.

MDAR checklist

## Data availability

All raw sequencing data generated in this study have been deposited as FASTQ files in the NCBI Sequence Read Archive (SRA) under the BioProject accession number PRJNA1026154. Accession numbers for RNA-seq datasets used in this study are shown in *Supplementary file 2*.

The following dataset was generated:

| Author(s) | Year | Dataset title | Dataset URL | Database and Identifier |
|---|---|---|---|---|
| Kobayashi H | 2025 | Lineage-specific genomic imprinting driven by LTR-derived sequence activating after fertilization | https://www.ncbi.nlm.nih.gov/bioproject/PRJNA1026154 | NCBI BioProject, PRJNA1026154 |

The following previously published datasets were used:

| Author(s) | Year | Dataset title | Dataset URL | Database and Identifier |
|---|---|---|---|---|
| Lynch VJ, Wagner GP | 2011 | Transposon-mediated gene regulatory network rewiring contributed to the evolution of pregnancy in mammals | https://www.ncbi.nlm.nih.gov/geo/query/acc.cgi?acc=GSE30708 | NCBI Gene Expression Omnibus, GSE30708 |
| Hughes DA, Stoneking M | 2015 | Evaluating intra- and inter-individual variation in the human placental transcriptome | https://www.ncbi.nlm.nih.gov/geo/query/acc.cgi?acc=GSE66622 | NCBI Gene Expression Omnibus, GSE66622 |
| Armstrong DL, Wildman D | 2017 | Expression Profiling of Term Placenta in Viviparous Mammals by RNA-Seq | https://www.ncbi.nlm.nih.gov/geo/query/acc.cgi?acc=GSE79121 | NCBI Gene Expression Omnibus, GSE79121 |
| Liu J | 2015 | Identification of gene expression changes in rabbit uterus during embryo implantation | https://www.ncbi.nlm.nih.gov/geo/query/acc.cgi?acc=GSE76115 | NCBI Gene Expression Omnibus, GSE76115 |
| Mika KM, Lynch VJ | 2020 | Evolutionary transcriptomics implicates HAND2 in the origins of implantation and regulation of gestation length | https://www.ncbi.nlm.nih.gov/geo/query/acc.cgi?acc=GSE155170 | NCBI Gene Expression Omnibus, GSE155170 |
| Wang X, Clark A | 2011 | Profiling of differential allelic expression in mouse placenta from reciprocal crosses | https://www.ncbi.nlm.nih.gov/geo/query/acc.cgi?acc=GSE29690 | NCBI Gene Expression Omnibus, GSE29690 |
| Coleman SJ, MacLeod JN | 2013 | Analysis of Unannotated Equine Transcripts Identified by mRNA Sequencing | https://www.ncbi.nlm.nih.gov/geo/query/acc.cgi?acc=GSE46858 | NCBI Gene Expression Omnibus, GSE46858 |
| Rosenkrantz JL | 2020 | RNA-seq of human and rhesus placental samples | https://www.ncbi.nlm.nih.gov/bioproject/PRJNA649979 | NCBI BioProject, PRJNA649979 |
| Necsulea A, Soumillon M, Liechti A, Daish T, Zeller U, Baker J, Grutzner F, Kaessmann H, Warnefors M | 2014 | The evolution of lncRNA repertoires and expression patterns in tetrapods | https://www.ncbi.nlm.nih.gov/geo/query/acc.cgi?acc=GSE43520 | NCBI Gene Expression Omnibus, GSE43520 |
| Zou Z, Zhang C, Wang Q, Hou Z, Zhao H, Chen Z, Xie W | 2022 | Translatome and transcriptome co-profiling reveals a role of TPRXs in human zygotic genome activation | https://www.ncbi.nlm.nih.gov/geo/query/acc.cgi?acc=GSE197265 | NCBI Gene Expression Omnibus, GSE197265 |

*Continued on next page*

*Continued*

| Author(s) | Year | Dataset title | Dataset URL | Database and Identifier |
|---|---|---|---|---|
| Kai Y, Mei H, Kawano H, Nakajima N, Takai A, Kumon M, Inoue A, Yamashita N | 2022 | Transcriptomic signatures in trophectoderm and inner cell mass of human blastocysts grouped according to developmental potential | https://www.ncbi.nlm.nih.gov/geo/query/acc.cgi?acc=GSE205171 | NCBI Gene Expression Omnibus, GSE205171 |
| Gong D | 2015 | T-WGBS for rhesus monkey sperm | https://www.ncbi.nlm.nih.gov/geo/query/acc.cgi?acc=GSM1466810 | NCBI Gene Expression Omnibus, GSM1466810 |
| Gong D | 2015 | T-WGBS for rhesus monkey oocytes | https://www.ncbi.nlm.nih.gov/geo/query/acc.cgi?acc=GSM1466811 | NCBI Gene Expression Omnibus, GSM1466811 |
| Krueger F | 2020 | DNA methylation analysis on genome-wide scale was performed using germ cells or early embryos from porcine or bovine specie | https://www.ncbi.nlm.nih.gov/geo/query/acc.cgi?acc=GSE143849 | NCBI Gene Expression Omnibus, GSE143849 |
| Samborski A, Graf A, Krebs S, Kessler B, Bauersachs S | 2013 | Deep Sequencing of the Porcine Endometrial Transcriptome on Day 14 of Pregnancy | https://www.ncbi.nlm.nih.gov/geo/query/acc.cgi?acc=GSE43667 | NCBI Gene Expression Omnibus, GSE43667 |
| Roslin Institute University of Edinburgh | 2014 | Population Genomics of sheep | https://www.ncbi.nlm.nih.gov/sra/?term=ERR489144 | NCBI Sequence Read Archive, ERR489144 |
| Wageningen University | 2012 | Pig placenta day 113 of pregnancy RNA-seq | https://www.ncbi.nlm.nih.gov/sra/?term=SRR543893 | NCBI Sequence Read Archive, SRR543893 |
| Broad Institute | 2012 | Treatment of genetic screening of hypertriglyceridemia type I, III, and V | https://www.ncbi.nlm.nih.gov/sra/?term=SRR558617 | NCBI Sequence Read Archive, SRR558617 |

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
