## [Editor Report · eLife Assessment]

The authors analyses describe a novel mechanism by which a retrotransposon-derived LTR may be involved in genomic imprinting and demonstrate imprinting of the ZDBF2 locus in rabbits and Rhesus macaques using allele-specific expression analysis. This imprinting of the ZDBF2 locus correlates with transcription of GPR1-AS orthologs. The accompanying genomic analysis is very well executed allowing for the conclusions reached in the manuscript. The revisions made at the request of the reviewers in this **important** manuscript strengthen the evidence from the genomic analyses, and as a result, the evidence is now **convincing** and will be informative to the genomics and developmental biology communities.

---

## [Referee Report · Reviewer #1 (Public review)]

Summary:

The study tests the conservation of imprinting of the ZBDF2 locus across mammals. ZDBF2 is known to be imprinted in mouse, human and rat. The locus has a unique mechanism of imprinting: although imprinting is conferred by a germline DMR methylated in oocytes, the DMR is upstream to ZDBF2 (at GPR1) and monoallelic methylation of the gDMR does not persist beyond early developmental stages. Instead, a lncRNA (GPR1-AS, also known as Liz in mouse) initiating at the gDMR is expressed transiently in embryos and sets up a secondary DMR (by mechanisms not fully elucidated) that then confers monoallelic expression of ZDBF2 in somatic tissues.

In this study, the authors first interrogate existing placental RNA-seq datasets from multiple mammalian species, and detect GPR1-AS1 candidate transcripts in human, baboon, macaque and mouse, but not in about a dozen other mammals. Because of the varying depth, quality and nature of these RNA-seq libraries, the ability to definitely detect the GPR1-AS1 lncRNA is not guaranteed; therefore, they generate their own deep, directional RNA-seq data from tissues/embryos from five species, finding evidence of GPR1-AS in rabbit, chimpanzee, but not bovine, pig or opossum. From these surveys, the authors conclude that the lncRNA is present only in Euarchontoglires mammals. To test the association between GPR1-AS and ZDBF2 imprinting, they perform RT-PCR and sequencing in tissue from wallabies and cattle, finding biallelic expression of ZDBF2 in these species that also lack a detected GPR1-AS transcript. From inspection of the genomic location of the GPR1-AS first exon, the authors identify an overlap with a solo LTR of the MER21C retrotransposon family in those species in which the lncRNA is observed, except for some rodents, including mouse. However, they do detect a degree of homology (46%) to the MER21C consensus at the first exon on Liz in mouse. Finally, the authors explore public RNA-seq datasets to show that GPR1-AS is expression transiently during human preimplantation development, an expression dynamic that would be consistent with the induction of monoallelic methylation of a somatic DMR at ZDBF2 and consequent monoallelic expression.

Strengths:

The analysis uncovers a novel mechanism by which a retrotransposon-derived LTR may be involved in genomic imprinting.

The genomic analysis is very well executed.

New directional and deeply-sequenced RNA-seq datasets from placenta or trophectoderm of five mammalian species and marsupial embryos, which will be of value to the community.

Weaknesses:

Although the genomic analysis is very strong, the study remains entirely correlative. All of the data are descriptive, and much of the analysis is performed on RNA-seq and other datasets from the public domain; a small amount of primary data is generated by the authors.

Evidence that the residual LTR in mouse is functionally relevant for Liz lncRNA expression is lacking.

Comments on revision:

The authors have responded very constructively to all points raised by me and the other reviewers. For example, the authors have gone to further, extensive efforts in seeking to identify an LTR at the mouse Liz locus - which is not found - but additional multiple genome alignments provide evidence for sequence conservation consistent with retention of a functional relic of the MER21C in rodent genomes. Moreover, they demonstrate the promoter activity of this mouse sequence region in transfections. They have also demonstrated imprinted expression of ZDBF2 in two additional species - rabbit and rhesus macaque - consistent with their model.

---

## [Referee Report · Reviewer #2 (Public review)]

Summary:

This work concerns the evolution of ZDBF2 imprinting in mammalian species via initiation of GPR1 antisense (AS) transcription from a lineage-specific long-terminal repeat (LTR) retrotransposon. It extends previous work describing the mechanism of ZDBF2 imprinting in mice and humans by demonstrating conservation of GPR1-AS transcripts in rabbits and non-human primates. By identifying the origin of GPR1-AS transcription as the LTR MER21C, the authors claim to account for how imprinting evolved in these species but not in those lacking the MER21C insertion. This illustrates the principle of LTR co-option as a means of evolving new gene regulatory mechanisms, specifically to achieve parent-of-origin allele specific expression (imprinting). Examples of this phenomenon have been described previously, but usually involve initiation of transcription during gametogenesis rather than post-fertilization, as in this work. The findings of this paper are therefore relevant to biologists studying imprinted genes or interested more generally in the evolution of gene regulatory mechanisms.

Strengths:

(1) The authors convincingly demonstrate the existence of GPR1-AS orthologs in specific mammalian lineages using high quality RNA-seq libraries collected from diverse mammalian species.

(2) The authors demonstrate imprinting of the ZDBF2 locus in rabbits and Rhesus macaques using allele-specific expression analysis. The transcription of GPR1-AS orthologs therefore correlates with imprinting of the ZDBF2 locus.

Weaknesses:

(1) Experimental evidence directly linking GPR1-AS transcription to ZDBF2 imprinting in rabbits and non-human primates is lacking. Consideration should be given to the challenges associated with studying non-model species and manipulating repeat sequences. Further, this mechanism is established in humans and mice, so the authors' model is arguably sufficiently supported merely by the existence of GPR1-AS orthologs in other mammalian lineages.

---

## [Referee Report · Reviewer #3 (Public review)]

Kobayashi et al identify MER21C as a common promoter of GPR1-AS/Liz in Euarchontoglires, which establishes a somatic DMR that controls ZFDB2 imprinting. In mice, MER21C appears to have diverged significantly from its primate counterparts and is no longer annotated as such.

The authors used high-quality cross-species RNA-seq data to characterise GPR1-AS-like transcripts, which included generating new data in five different species. The association between MER21C/B elements and the promoter of GPR1-AS in most species is clear and convincing. The expression pattern of MER21C/B elements overall further supports their role in enabling correct temporal expression of GPR1-AS during embryonic development.

In the revised version of the manuscript the authors provided additional support for the common evolutionary origin of the GPR1-AS/Liz promoter between primates and rodents. They also showed a more extensive concordance between the presence of GPR1-AS-like transcripts and ZDBF2 imprinting.

Altogether, these findings robustly support the conclusions of the paper, shedding light into the events underlying the evolution of imprinting at the ZDBF2 locus.

---

## [Author Response]

The following is the authors’ response to the original reviews

**Recommendations For The Authors:**

**Reviewer #1 (Recommendations For The Authors):**
Recommendations Analysis:(1) Given that a MER21B/C LTR was not immediately identified at the start site of the Liz lncRNA in the mouse, and its match is only 46%, this raises the question of whether an analogous LTR would be identified at the homologous location in other species on deeper analysis. The authors need to argue that what has been conserved in the LTR alone in mouse is the essential element conferring the ability to initiate transcription of Liz. A transient reporter assay might be sufficient to do this.

We believe that the 46% identity between the first exon of mouse Liz and the consensus sequence of MER21C is so weak that its traces as MER21C are too attenuated to be detected by standard in silico analyses, such as homology searches. For instance, when pairwise alignments are performed between the first exon of mouse Liz and the consensus sequences of solo-LTRs other than MER21C, MER21C does not emerge as the most similar sequence (Figure 5 – figure supplement 1). This is in stark contrast to similar analyses involving the first exon of human and rabbit GPR1AS (which overlaps with MER21C), where MER21C is identified as the most similar sequence. [pages: 26, 31-32]

The positions of these LTRs were initially annotated using RepeatMasker. To ensure robust analysis, we performed additional searches with RepeatMasker under more sensitive conditions, adjusting search engines (e.g., RMblast to HMMER or Cross-match) and sensitivity settings. Nevertheless, MER21C or closely related LTRs were still undetectable in mouse, rat, and hamster (Figure 4 – figure supplement 1). However, a multiple genome alignment generated by Cactus/UCSC revealed a syntenic region corresponding to the first exon of human GPR1-AS, overlapping with LTR21C, in the genomes of mice, as well as rats and hamsters (Figure 4 – figure supplement 2). Although RepeatMasker did not annotate MER21C at the GPR1 locus in these species, homologous regions were observed across all selected Euarchontoglires. Due to the limitations of the Cactus alignment track in delineating precise homologous boundaries across species, extracting sequences for evolutionary tree construction was not feasible. Nevertheless, these findings support the hypothesis that the first exon of GPR1-AS (Liz in mice) originated from a MER21C insertion in the common ancestor of Euarchontoglires. [pages: 21, 24-25]

A combination of traditional annotation of repetitive elements using RepeatMasker and the reconstruction of ancestral genomes through multiple genome alignment can reveal highly degenerated LTR relics. This approach is likely to point to significant future directions for research. This point is further elaborated in the discussion section. [page 42]

Furthermore, in response to the reviewer's suggestion, we investigated the promoter activity of the GPR1-AS and Liz first exons, which are hypothesized to have originated from the same MER21C insertion. Using a dual reporter assay, we demonstrated that the first exon of mouse Liz exhibits promoter activity in a human cell line comparable to that of the human GPR1-AS promoter. Thus, despite the relatively low sequence similarity between the Liz first exon and the MER21C consensus sequence (46% as determined by pairwise alignment, Figure 5 – figure supplement 2), the promoter activity remains functionally conserved. We further discuss the potential functional motifs within the putative MER21C LTR-derived sequences in Figure 4B-D. Taken together, these findings suggest that despite a high level of degeneracy of the promoter region in rodents, including mice, the most parsimonious explanation for the origin of this regulatory element in rodents is the presence of the same LTR relic detectable in humans/primates, which is essential for robust transcription initiation of Liz and GPR1-AS, respectively. [pages: 27, 32]

(2) Imprinting will depend on an initiating mechanism in the germline, in addition to events in the embryo that induce the secondary DMR at ZDBF2. The authors should therefore examine as far as possible the presence of a gDMR in the species with/without GPR1-AS1 and ZDBF2 imprinting. Whole-genome bisulphite sequencing data from oocytes and sperm should exist for some of the relevant species (e.g., pig, cow: Ivanova et al. 2020 PMID: 32393379; Lu et al. 2012 PMID: 34818044).

As the reviewer noted, the presence of a gDMR is essential for the establishment of imprinting. Following another reviewer's suggestion, we have now demonstrated that the ZDBF2 gene in rhesus monkeys is also subject to imprinting (see Figure 3C-D). We also acquired whole genome bisulfite sequencing data for rhesus monkey sperm and oocytes, identified DMRs between them, and discovered an oocyte-specifically methylated gDMR in the first exon of GPR1-AS (which overlaps with MER21C)(Figure 3 – figure supplement 1A). This finding is consistent with observations in humans and mice. Conversely, we obtained similar sequencing data for porcine and bovine sperm and oocytes and conducted the same analysis (Figure 3 – figure supplement 1A,B). However, we did not detect any oocyte-specific methylated gDMRs in the GPR1 intragenic region (where GPR1AS is transcribed from an intron of GPR1) in these species of the Laurasiatheria superorder. These results support the hypothesis that ZDBF2 is not imprinted in lineages outside the Euarchontoglires, the superorder which includes both rodents and primates. We have included these important DMR results as a supplement to Figure 3. [pages 16-21]

Presentation:(1) The first section of the Introduction would benefit from the inclusion of some additional general references on genomic imprinting.

We have added two review articles, Tucci et al. (2019) and Kobayashi (2021), as references in the first section of the Introduction. [page 5]

(2) Introduction statement: "....nearly 200 imprinted genes have been identified in mice and humans. However, less than half of these genes overlapped in both species." This was the conclusion of one study (Tucci et al. 2016), so it would be better to provide a caveat to the statement "However, one comparative analysis suggested that fewer than half of these genes overlapped in both species".The point being that the actual number of imprinted genes is still a matter of debate (see Edwards et al. 2023 PMID: 36916665), and the extent of overlap will depend on the strength of the evidence for each gene in the human and mouse imprinted gene lists. So, it is very difficult to put an accurate figure on the extent of overlap - but the authors' point is valid that there are species- or lineage-specific imprinted genes.

We have revised this sentence following reviewer #1's suggestion. [page 5]

(3) Introduction statement: "The establishment of species-specific imprinting.....can be driven by various evolutionary events, including.....differences in the function of DNA methyltransferases". I am not aware that this has been described as an evolutionary event causing species-specific imprinting - without supporting evidence, I recommend to remove this suggestion.

We thank the reviewer for this comment and realize that we should have been more explicit here. We were referring to DNMT3C, a rodent-specific member of the DNMT3 family, which is responsible for the paternal methylation imprinting of Rasgrf1 in mice (Barau et al., Science, 2016), in association with the piRNA pathway and targeting of a specific retrotransposon within the DMR (Watanabe et al. Science, 2011). The Rasgrf1 gene is imprinted in mice but not considered imprinted in humans (though some conflicting data exist). While it is likely that the emergence of DNMT3C was a pre-requisite to the establishment of Rasgrf1 imprinting in evolutionary terms, clear evidence is lacking. Following the reviewer’s suggestion, we have removed the phrase "differences in the function of DNA methyltransferases" from the text. However, we have reintroduced this point in the Introduction section as a potential mechanism that may contribute to the establishment of species-specific imprinted genes, alongside the roles of ZNF445 and ZFP57, which regulate the maintenance of imprinting with partially divided roles between humans and mice. [page 6]

(4) It would be very useful for readers to have a schema of the Gpr1/Zdbf2 locus that indicates the locations of the germline and somatic DMRs and their relationship to the Liz transcript.(5) There is a summary figure amongst the Supplementary Figures (Suppl. Fig. 7) - it would be beneficial to readers to have this summary figure in the main text rather than the supplement.

Following reviewer #1’s suggestion, we have moved the regulatory system schema at the Gpr1/Zdbf2 locus, originally shown in Supplementary Figure 7, to the main text as Figure 7. In addition, in response to comment 4, we have revised the figure to explicitly depict the relationship between the Liz transcript and the establishment of the somatic DMR (sDMR), enhancing the clarity of the regulatory interactions at this locus. [page 38]

(6) With a focus of the study on LTRs as cis-regulatory elements having been co-opted in genomic imprinting mechanisms - whether in the female germline (as in Bogutz et al. 2019) or in the current study as an activating element post-fertilisation - it is a real omission that the authors do not to refer to the role of tissue-specific LTRs as the candidate regulatory elements in non-canonical imprinting (see Hanna et al. 2019 PMID: 31665063). Please include in Introduction and/or Discussion.

We added a sentence explaining canonical and non-canonical imprinting and the cases where LTRs act as regulatory elements in non-canonical imprinting, with reference to the study of Hanna et al., as suggested. [page 6]

(7) Discussion statement: "Two paternally expressed imprinted genes, PEG10/SIRH1 and PEG11/RTL1/SIRH2 have been identified in mammals. They encode GAG-POL proteins of sushi-ichi LTR retrotransposons and are essential for mammalian placenta formation and maintenance."These sentences should be combined: "Two paternally expressed imprinted genes, PEG10/SIRH1, and PEG11/RTL1/SIRH2, that encode GAG-POL proteins of sushi-ichi LTR retrotransposons have been identified in mammals and are essential for mammalian placenta formation and maintenance."

We have revised this sentence according to reviewer #1's suggestion. [page 41]

**Reviewer #2 (Recommendations For The Authors):**
When showing assembled GPR1-AS transcripts via genome browser tracks, it would be valuable to add normalized counts of reads mapping to each strand, in order to more convincingly demonstrate the existence of these transcripts. I ask for this because in my experience Stringtie will assemble transcripts that are only marginally supported by reads.

In response to Reviewer #2's suggestion, FPKM and TPM values for all StringTiepredicted GPR1-AS-like transcripts are now included in Figure 6. Each of these transcripts has a TPM value greater than 1, supporting their validity. [pages: 35]

**Reviewer #3 (Recommendations For The Authors):**
(1) The tree in Figure 5A is one of the main arguments supporting the divergence of the mouse Liz promoter from a common MER21C element, but this contains only a handful of species, making it difficult to appreciate the full extent of its evolution. Presumably its faster mutation rate in mouse would also be supported by other closely related rodents, which would solidify the conclusion that the Liz promoter is derived from an ancient MER21C insertion. So my suggestion is to expand this tree substantially to other species, comparing sequences syntenic to the GPR1-AS/Liz promoter.(2) It may also be worth trying different TE/LTR annotation tools and/or running Repeatmasker with different parameters, to see if an MER21C element is detected in mouse using a more sensitive approach.

In response to this suggestion, we performed computational analyses with RepeatMasker under various settings (e.g., switching search engines from RMblast to HMMER or Crossmatch, adjusting speed/sensitivity settings from default to slow). Despite these modifications, a MER21C element was not detected near the mouse Liz promoter. However, a multiple genome alignment track generated by Cactus/UCSC revealed a syntenic region, corresponding to the first exon of human GPR1-AS, which overlaps with LTR21C, also present in the genomes of mouse, rat, and hamster (Figure 4 – figure supplement 1). While RepeatMasker did not identify MER21C at the GPR1 locus in these species, homologous regions were observed across all selected Euarchontoglires. Although the Cactus alignment track does not delineate the exact boundaries of homologous regions across species (relative to humans) and thus precludes extracting each homologous sequence to construct an evolutionary tree, these findings support the hypothesis that the first exon of GPR1-AS (referred to as Liz in mice) originated from an ancient MER21C insertion in the common ancestor of Euarchontoglires. [pages: 21, 24-25]

(3) According to Dfam, MER21C is not common to all eutherians, but specific to Boroeutheria, whilst MER21B is presumably specific to Euarchontoglires. To clarify MER21C/B evolution, it would be useful to show the number of elements present in select species from each group (including an outgroup).(7) In Figure 4 it is hard to distinguish between red and purple.

Initially, we referenced Repbase (e.g., MER21C: Origin/Eutheria), but, as Reviewer #3 noted, Dfam should be the primary reference. We have now included the copy numbers of MER21C and MER21B for each genome in Figure 4, providing a clearer understanding of their evolutionary appearance (MER21C appears specific to Boroeutheria, while MER21B is specific to Euarchontoglires). Additionally, we adjusted the MER21B position color from purple to dark purple to improve visibility. Furthermore, we have also underlined the copy number of MER21C or MER21B located within the GPR1 region in each species. For example, in the Treeshrew genome, the LTR overlapping with GPR1-AS is annotated as MER21B, so we underlined the copy number of MER21B (2,305). These changes now clearly indicate whether homologous sequences to the first exon of GPR1-AS are annotated as MER21C or MER21B in each genome. [page 22]

(4) Could the imprinting status of ZDBF2 not be determined in chimpanzees and rabbits? Or is it already known? Either way, a clarification would be useful to further support the concordance between GPR1-AS-like transcripts and ZDBF2 imprinting.

The imprinting status of ZDBF2 had not previously been reported in chimpanzees, rhesus macaques, or rabbits, where GPR1-AS-like transcripts were identified. Therefore, we conducted allele-specific expression analysis of ZDBF2 using blood samples from rhesus macaques and rabbits. As expected, paternal-allele-specific expression of ZDBF2 was observed in both species, consistent with findings in humans and mice. These results have been added to Figure 3. Although we did not analyze the imprinting status in chimpanzees, we believe the existing data sufficiently support our hypothesis. [pages: 16, 19-20]

(5) The authors briefly discuss the role of KRAB-ZFPs in controlling TE expression. An interesting addition would be to analyse the expression of the main KRAB-ZFP that binds to MER21C (ZFP789, according to data from PMID 28273063). This could be linked to the temporal control of MER21C expression.

In response to Reviewer #3's suggestion, we focused on the expression pattern of ZNF789 (noted by the reviewer as ZFP789), the primary KRAB-ZFP known to bind MER21C, as identified by Didier Trono’s group (PMID 28273063). Strikingly, our analysis reveals that ZNF789 is specifically downregulated at the 4-cell stage, which aligns with the timing of MER21C reactivation. While it remains to be determined whether this downregulation directly influences MER21C reactivation or the initiation of GPR1-AS expression, this finding is significant and consistent with our model. We have incorporated this information in Figure 5 – figure supplement 3. [pages: 33]

(6) The sentence "Liz directs DNA methylation at the somatic DMR, which competes with ZDBF2 to repress the paternal allele" (introduction) was confusing to me.

This sentence has been revised to be more accurate as follows; Liz transcription counteracts the H3K27me3-mediated repression of Zdbf2 by promoting the deposition of antagonistic DNA methylation at the secondary DMR. [page 7]

(8) In Figure 5 I take it that 'consensus motif' refers to ELF1/2? Maybe change the legend.

To clarify potential confusion around the term 'consensus motif,' which may have been mistaken for 'consensus MER21C' (the consensus sequence of MER21C-LTR from the Dfam database), we have revised the figure legend. We now refer to the motif as the "common motif," indicating the sequence common to all MER21C-derived sequences and overlapping with the first exon of GPR1-AS. [page 29]